# A geometric attractor mechanism for self-organization of entorhinal grid modules

**Louis Kang[1,2]\*, Vijay Balasubramanian[1]**

[1]David Rittenhouse Laboratories, University of Pennsylvania, Philadelphia, United States; [2]Redwood Center for Theoretical Neuroscience, University of California, Berkeley, Berkeley, United States

**Abstract** Grid cells in the medial entorhinal cortex (MEC) respond when an animal occupies a periodic lattice of 'grid fields' in the environment. The grids are organized in modules with spatial periods, or scales, clustered around discrete values separated on average by ratios in the range 1.4–1.7. We propose a mechanism that produces this modular structure through dynamical self-organization in the MEC. In attractor network models of grid formation, the grid scale of a single module is set by the distance of recurrent inhibition between neurons. We show that the MEC forms a hierarchy of discrete modules if a smooth increase in inhibition distance along its dorso-ventral axis is accompanied by excitatory interactions along this axis. Moreover, constant scale ratios between successive modules arise through geometric relationships between triangular grids and have values that fall within the observed range. We discuss how interactions required by our model might be tested experimentally.
DOI: https://doi.org/10.7554/eLife.46687.001

## Introduction

A grid cell has a spatially modulated firing rate that peaks when an animal reaches certain locations in its environment (*Hafting et al., 2005*). These locations of high activity form a regular triangular grid with a particular length scale and orientation in space. Every animal has many grid cells that collectively span a wide range of scales, with smaller scales enriched dorsally and larger scales ventrally along the longitudinal axis of the MEC (*Stensola et al., 2012*). Instead of being smoothly distributed, grid scales cluster around particular values and thus grid cells are partitioned into modules (*Stensola et al., 2012*). Consecutive pairs of modules have scale ratios in the range 1.2–2.0 (*Stensola et al., 2012*; *Barry et al., 2007*; *Krupic et al., 2015*). The scale ratio averaged across animals is constant from one pair of modules to the next and lies in the interval 1.4 (*Stensola et al., 2012*) to 1.7 (*Barry et al., 2007*; *Krupic et al., 2015*), suggesting that the grid system favors a universal scale ratio in this range.

Encoding spatial information through grid cells with constant scale ratios is thought to provide animals with an efficient way of representing their position within an environment (*Moser et al., 2008*; *Fiete et al., 2008*; *Mathis et al., 2012*; *Wei et al., 2015*; *Stemmler et al., 2015*; *Sanzeni et al., 2016*; *Mosheiff et al., 2017*). Moreover, periodic representations of space permit a novel mechanism for precise error correction against neural noise (*Sreenivasan and Fiete, 2011*) and are learned by machines seeking to navigate open environments (*Cueva and Wei, 2018*; *Banino et al., 2018*). These findings provide motivation for forming a modular grid system with a constant scale ratio, but a mechanism for doing so is unknown. Continuous attractor networks (*Fuhs and Touretzky, 2006*; *Burak and Fiete, 2009*), a leading model for producing grid cells, would currently require discrete changes in scales to be directly imposed as sharp changes in parameters, as would the oscillatory interference model (*Burgess et al., 2007*; *Hasselmo et al., 2007*) or hybrid models (*Bush and Burgess, 2014*). In contrast, many sensory and behavioral systems have

**\*For correspondence:**
louis.kang@berkeley.edu

**Competing interests:** The authors declare that no competing interests exist.

**eLife digest** In a room, we have a sense of our location relative to the doors and to objects within the room. This is because the brain constructs a mental map of our current environment. As we move around the room, neurons called grid cells fire whenever we are in specific locations. But these locations are not random. They correspond to the corners of a grid of tessellating triangles on the floor, a little like the dots in a regular polka-dot pattern. Grid cells fire whenever we stand on one of the dots. This enables the brain to keep track of where we are and where we are heading.

But the brain does not use just a single grid cell map to represent a room. Instead, it uses multiple maps with different spatial scales. These maps differ in the distance between the points at which each grid cell fires, that is, the distance between the polka dots. Some maps have many small triangles, providing high resolution spatial information. Others have fewer, larger triangles. This is similar to how we use maps with different spatial scales when driving between cities versus walking around a single neighborhood. A set of grid cell maps with the same spatial scale—and the same orientation—is known as a grid cell module.

Animal experiments suggest that different individuals use a similar combination of grid cell modules that can efficiently map rooms. But how can the brain reliably produce this particular combination? Using a computer model to simulate networks of grid cells, Kang and Balasubramanian identify a mechanism that enables the brain to spontaneously organize into the previously observed combination. The interactions between networks—in particular the balance of inhibitory and excitatory activity—determine the arrangement of grid cell modules. This process still works even with random fluctuations in network activity.

Grid cells occupy a brain region that degenerates early in the course of Alzheimer's disease. This may explain why some patients experience difficulty finding their way around as one of their first symptoms. To develop effective treatments, scientists need to understand how neural circuits within this brain region work, and how the disease process disrupts them. The computer model of Kang and Balasubramanian brings the research community a step closer to achieving this. It also provides insights into how neuronal networks self-organize, which is relevant to other brain functions too.

DOI: https://doi.org/10.7554/eLife.46687.002

smooth tuning distributions, such as preferred orientation in visual cortex (*Issa et al., 2008*) and preferred head direction in the MEC (*Taube et al., 1990*). A self-organizing map model with stripe cell inputs (*Grossberg and Pilly, 2012*) and a firing rate adaptation model with place cell inputs (*Urdapilleta et al., 2017*) can generate discrete grid scales, but their ratios are not constant or constant-on-average unless explicitly tuned.

Here, we present a simple extension of the continuous attractor model that adds excitatory connections between a series of attractor networks along the dorso-ventral axis of the MEC, accompanied by an increase in the distance of inhibition. The inhibition gradient drives an increase in grid scale along the MEC axis. Meanwhile, the excitatory coupling discourages changes in grid scale and orientation unless they occur through geometric relationships with defined scale ratios and orientation differences. Competition between the effects of longitudinal excitation and lateral inhibition self-organizes the complete network into a discrete hierarchy of modules. Certain grid relationships are geometrically stable, which makes them, and their associated scale ratios, insensitive to perturbations. The precise ratios that appear depend on the balance between excitation and inhibition and how it varies along the MEC axis. We show that sampling across a range of these parameters leads to a distribution of scale ratios that matches experiment and is, on average, constant from the smallest to the largest pair of modules.

Continuous attractors are a powerful general method for self-organizing neural dynamics. To our knowledge, our results are the first demonstration of a mechanism for producing a discrete hierarchy of modules in a continuous attractor system.

## Results

### Standard grid cell attractors are not modular

We assemble a series of networks along the longitudinal MEC axis, numbering them $z$ = 1, 2, ..., 12 from dorsal to ventral (*Figure 1A*). Each network contains the standard 2D continuous attractor architecture of the Burak-Fiete model (*Burak and Fiete, 2009*). Namely, neurons are arranged in a 2D sheet with positions (*x,y*), receive broad excitatory drive (*Bonnevie et al., 2013* and *Figure 1B*), and inhibit one another at a characteristic separation on the neural sheet (*Figure 1C*; see Materials and methods for a complete description). In our model, this inhibition distance $l$ is constant within each network but increases from one network to the next along the longitudinal axis of the MEC. With these features alone, the population activity in each network self-organizes into a triangular grid whose lattice points correspond to peaks in neural activity (*Figure 2A*). Importantly, the scale of each network's grid, which we call $\lambda(z)$, is proportional to that network's inhibition distance $l(z)$ ('uncoupled' simulations in *Figure 3A*). Also, network grid orientations θ show no consistent pattern across scales and among replicate simulations with different random initial firing rates.

Following the standard attractor model (*Burak and Fiete, 2009*), the inhibitory connections in each network are slightly modulated by the animal's velocity such that the population activity pattern of each network translates proportionally to animal motion at all times (Materials and methods). This modulation allows each network to encode the animal's displacement through a process known as path-integration, and projects the network grid pattern onto spatial rate maps of single neurons. That is, a recording of a single neuron over the course of an animal trajectory would show high activity in spatial locations that form a triangular grid with scale Λ (*Figure 2C*). Moreover, Λ(z) for a neuron from network $z$ is proportional to that network's population grid scale $\lambda(z)$, and thus also proportional to its inhibition distance $l(z)$ (uncoupled simulations in *Figure 3B*). To be clear, we call Λ the 'spatial scale'; it corresponds to a single neuron's activity over the course of a simulation and has

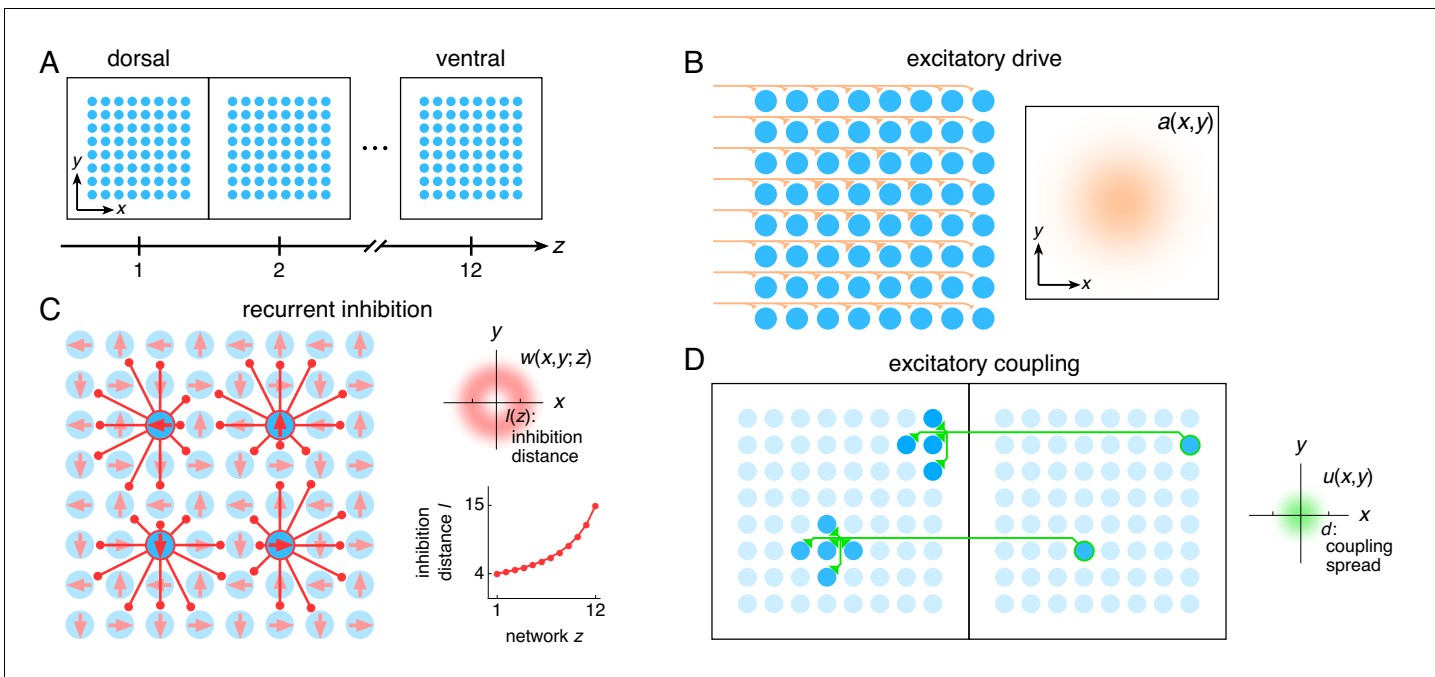

**Figure 1.** The entorhinal grid system as coupled 2D continuous attractor networks (Materials and methods). (**A**) Each network $z$ corresponds to a region along the dorso-ventral MEC axis and contains a 2D sheet of neurons with positions (*x,y*). (**B**) Neurons receive excitatory drive *a(x,y)* that is greatest at the network center and decays toward the edges. (**C**) Neurons inhibit neighbors within the same network with a weight *w(x,y;z)* that peaks at a distance of *l(z)* neurons, which increases as a function of $z$. Each neuron has its inhibitory outputs shifted slightly in one of four preferred network directions and receives slightly more drive when the animal moves along its preferred spatial direction. (**D**) Each neuron at position (*x,y*) in network $z$ excites neurons located within a spread *d* of (*x,y*) in network $z$ – 1.
DOI: https://doi.org/10.7554/eLife.46687.003

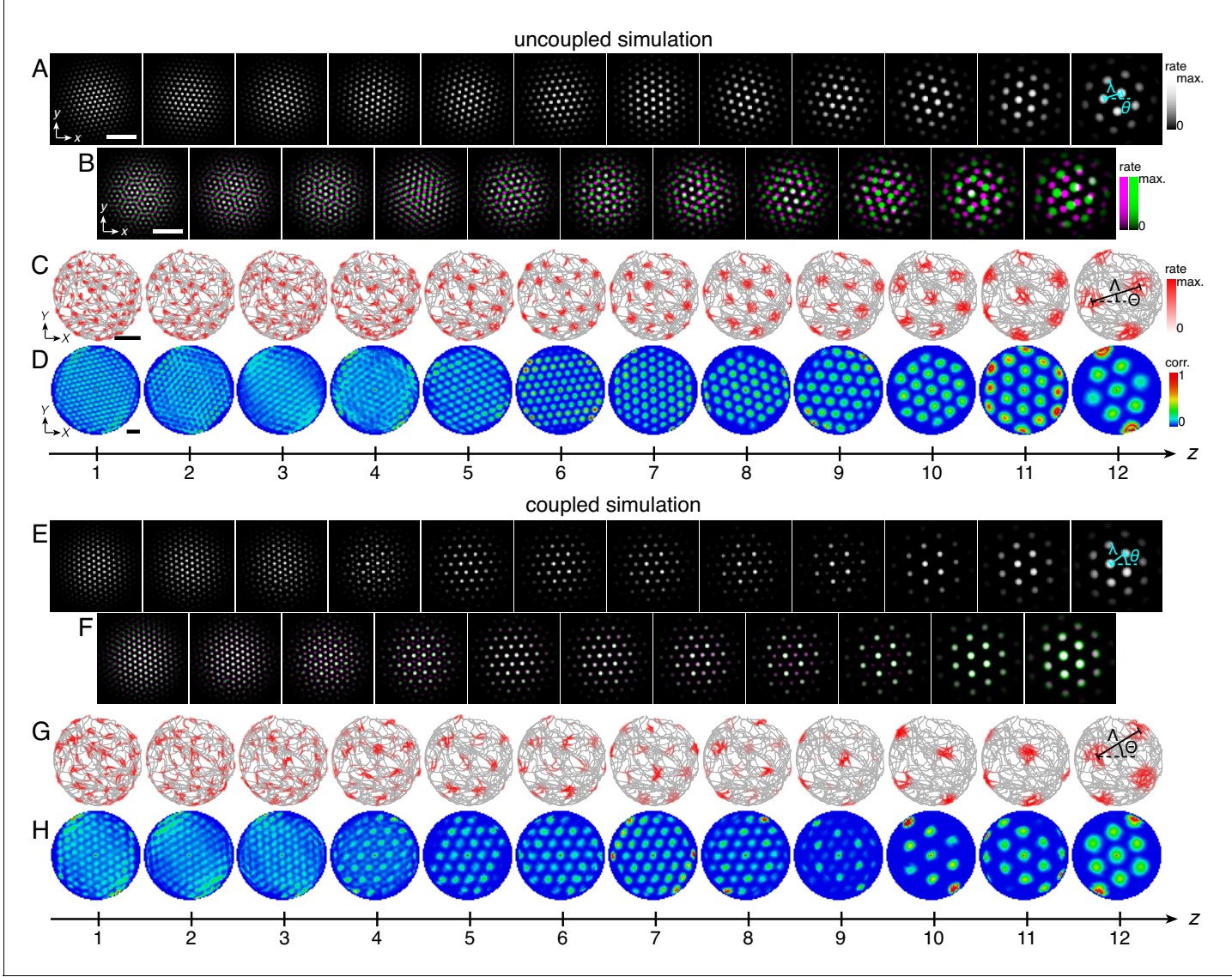

**Figure 2.** Uncoupled and coupled systems produce grid cells with a range of scales. (A–D) A representative simulation without coupling. (A) Network activities at the end of the simulation. (B) Activity overlays between adjacent networks depicted in A. In each panel, the network with smaller (larger) z is depicted in magenta (green), so white indicates activity in both networks. (C) Spatial rate map of a single neuron for each z superimposed on the animal's trajectory. (D) Spatial autocorrelations of the rate maps depicted in C. (E–H) Same as A–D but for a representative simulation with coupling. Standard parameter values provided in **Table 1**. White scale bars, 50 neurons. Black scale bars, 50 cm.

DOI: https://doi.org/10.7554/eLife.46687.004

units of physical distance in space. By contrast, $\lambda$, the 'network scale' described above, corresponds to the population activity at a single time and has units of separation on the neural sheet. Similarly, $\Theta(z)$ describes the orientation of the spatial grid of a single neuron in the network $z$; we call $\Theta$ the 'spatial orientation.' Like the network orientations $\theta$ discussed above, spatial orientations of grids show no clustering (uncoupled simulations in *Figure 3B*).

With an inhibition distance $l(z)$ that increases gradually from one network to the next (*Figure 1C*), proportional changes in network and spatial scales $\lambda(z)$ and $\Lambda(z)$ lead to a smooth distribution of grid scales (uncoupled simulations in *Figure 3A,B*). To reproduce the experimentally observed jumps in grid scale between modules, the inhibition distance would also have to undergo discrete, sharp jumps between certain adjacent networks. In summary, a grid system created by disjoint attractor networks will not self-organize into modules.

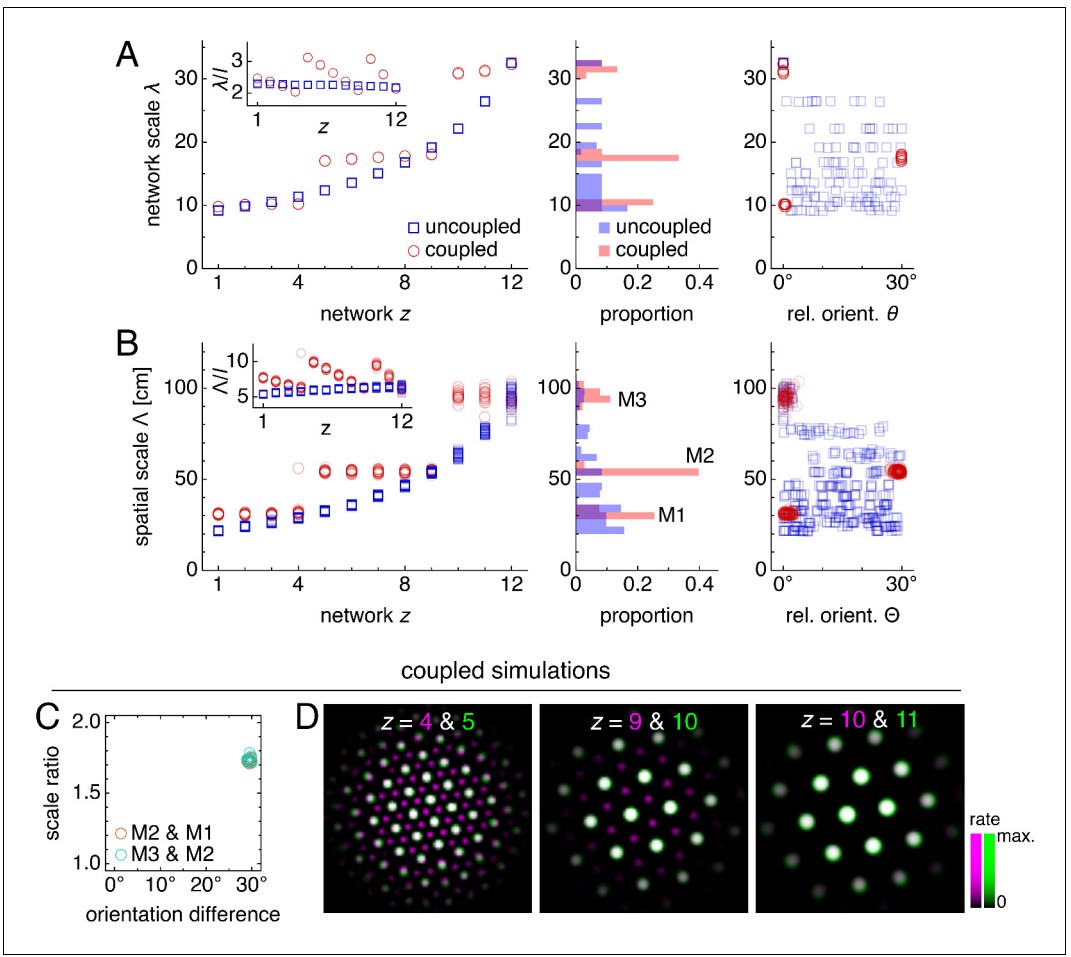

**Figure 3.** Coupling can induce modularity with fixed scale ratios and orientation differences. (**A–C**) Data from 10 replicate uncoupled and coupled simulations. (**A**) Left: network grid scales $\lambda(z)$. For each network, there are 10 closely spaced red circles and 10 closely spaced blue squares corresponding to replicate simulations. Inset: $\lambda(z)$ divided by the inhibition distance $l(z)$. Middle: histogram for $\lambda$ collected across all networks. Right: network grid orientations $\theta$ relative to the network in the same simulation with largest scale. (**B**) Left: spatial grid scales $\Lambda(z)$. For each $z$, there are up to 30 red circles and 30 blue squares corresponding to three neurons recorded during each simulation. Inset: $\Lambda(z)$ divided by the inhibition distance $l(z)$. Middle: histogram for $\Lambda$ collected across all networks. In the coupled model, grid cells are clustered into three modules. Right: spatial grid orientations $\Theta$ relative to the grid cell in the same simulation with largest scale. (**C**) Spatial scale ratios and orientation differences between adjacent modules for the coupled model. (**D**) Activity overlays enlarged from *Figure 2F* to emphasize lattice relationships. In each panel, the network with smaller (larger) $z$ is depicted in magenta (green), so white indicates activity in both networks. Standard parameter values provided in *Table 1*.
DOI: https://doi.org/10.7554/eLife.46687.005

## Coupled attractor networks produce modules

Module self-organization can be achieved with one addition to the established features listed above: we introduce excitatory connections from each neuron to those in the preceding network with approximately corresponding neural sheet positions (*Figure 1D*; see Materials and methods for a complete description). That is, a neuron in network $z$ (more ventral) with position $(x,y)$ will excite neurons in network $z - 1$ (more dorsal) with positions that are within a distance $d$ of position $(x,y)$. In other words, the distance $d$ is the 'spread' of excitatory connections, and we choose a constant value across all networks comparable to the inhibition distance $l(z)$.

The self-organization of triangular grids in the neural sheet and the faithful path-integration that projects these grids onto single neuron spatial rate maps persist after introduction of inter-network coupling (*Figure 2G*). Network and spatial scales $\lambda(z)$ and $\Lambda(z)$ still increase from network $z = 1$

(dorsal) to network $z = 12$ (ventral). Yet, *Figure 3A,B* shows that for the coupled model, these scales exhibit plateaus that are interrupted by large jumps, disrupting their proportionality to inhibition distance $l(z)$, which is kept identical to that of the uncoupled system (*Figure 1C*). Collecting scales across all networks illustrates that they cluster around certain values in the coupled system while they are smoothly distributed in the uncoupled system. We identify these clusters with modules M1, M2, and M3 of increasing scale. Note that multiple networks at various depths $z$ can belong to the same module. Moreover, coupling causes grid cells that cluster around a certain scale to also cluster around a certain orientation (*Figure 3A,B*), as seen in experiment (*Stensola et al., 2012*). The uncoupled system does not demonstrate co-modularity of orientation with scale, that is two networks with similar grid scales need not have similar orientations unless this is imposed by an external constraint.

In summary, excitatory coupling between grid attractor networks dynamically induces discreteness in grid scales that is co-modular with grid orientation, as observed experimentally (*Stensola et al., 2012*), and as needed for even coverage of space by the grid map (*Sanzeni et al., 2016*).

## Modular geometry is determined by lattice geometry

Not only does excitatory coupling produce modules, it can do so with consistent scale ratios and orientation differences. For the coupled system depicted in *Figure 2*, scale ratios and orientation differences between pairs of adjacent modules consistently take values 1.74 ± 0.02 and 29.5 ± 0.4°, respectively (mean ± s.d.; *Figure 3C*). These values are robust to a variety of parameter perturbations, coupling architectures, and sources of noise. We can make the inhibition distance profile $l(z)$ less or more concave (*Figure 4A,B*), or we can implement excitatory connections with different properties by reversing their direction (*Figure 4C*), including connections in both directions (*Figure 4D*), or allowing the coupling spread to vary with network depth (*Figure 4E*). In each case, the same scale ratio of ≈1.7 and orientation difference of ≈30° persist. We can also reduce the number of neurons by a factor of 9 without affecting the scale ratio and orientation difference (*Figure 4F*). Similar results are obtained with neural inputs corrupted by independent Gaussian noise (*Figure 4G*) and with randomly shifted excitatory connections, which adds another form of coupling imprecision in addition to spread (*Figure 4H*). Finally, simulations with spiking dynamics following *Burak and Fiete (2009)* also demonstrate a preference for scale ratios of ≈1.7 and orientation differences of ≈30°, albeit with greater variability (*Figure 4I*).

We can intuitively understand this robust modularity through the competition between lateral inhibition within networks and longitudinal excitation across networks. In the uncoupled system, grid scales decrease proportionally as the inhibition distance $l(z)$ decreases from $z = 12$ to $z = 1$. However, coupling causes areas of high activity in network $z$ to preferentially excite corresponding areas in network $z - 1$, which encourages adjacent networks to share the same grid pattern ($z = 10$ & 11 in *Figure 3D*). Thus, coupling adds rigidity to the system and provides an opposing 'force' against the changing inhibition distance that attempts to drive changes in grid scale. This rigidity produces the plateaus in network and spatial scales $\lambda(z)$ and $\Lambda(z)$ that delineate modules across multiple networks.

At interfaces between modules, coupling can no longer fully oppose the changing inhibition distance, and the grid pattern changes. However, the rigidity fixes a geometric relationship between the grid patterns of the two networks spanning the interface. In the coupled system of *Figure 2* and *Figure 3*, module interfaces occur between networks $z = 4$ and 5 and between $z = 9$ and 10. The network population activity overlays of *Figure 3D* reveal overlap of many activity peaks at these interfaces. However, the more dorsal network (with smaller $z$) at each interface contains additional small peaks between the shared peaks. In this way, adjacent networks still share many corresponding areas of high activity, as favored by coupling, but the grid scale changes, as favored by a changing inhibition distance. Pairs of grids whose lattice points demonstrate regular registry are called *commensurate* lattices (*Chaikin and Lubensky, 1995*) and have precise scale ratios and orientation differences, here respectively $\sqrt{3} \approx 1.7$ and 30°, which match the results in *Figure 3C* and *Figure 4*.

In summary, excitatory coupling can compete against a changing inhibition distance to produce a rigid grid system whose 'fractures' exhibit stereotyped commensurate lattice relationships. These robust geometric relationships lead to discrete modules with fixed scale ratios and orientation differences.

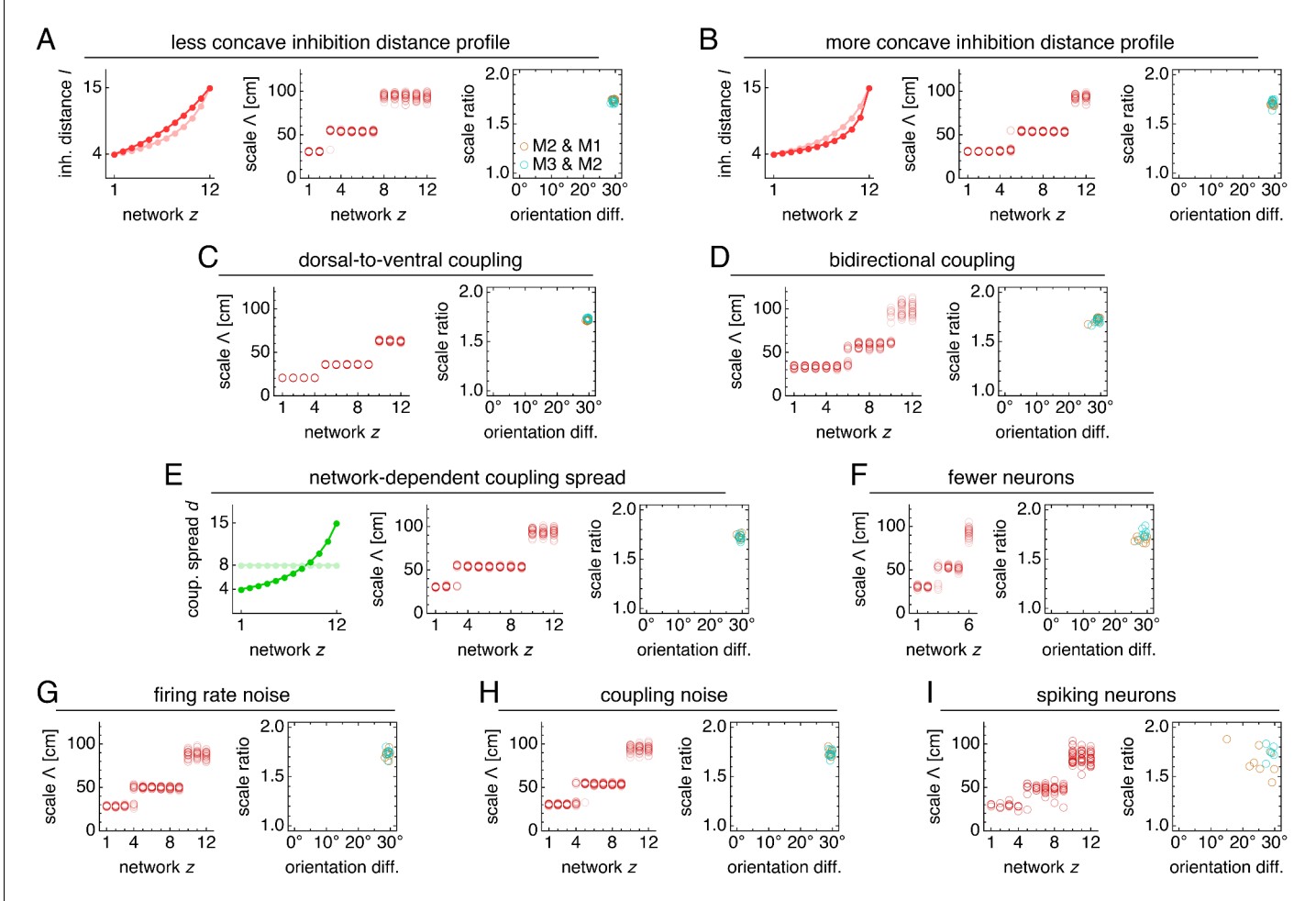

**Figure 4.** Modules produced by commensurate lattices maintain the same scale ratios and orientation differences across various perturbations, architectures, and sources of noise. Data from 10 replicate simulations in each subfigure, which shows spatial grid scales Λ(z) and scale ratios and orientation differences between modules. (**A**) Left: less concave inhibition distance profile l(z) (dark) compared to *Figure 1C* (light). (**B**) Same as **A**, but for a more concave l(z). (**C**) Dorsal-to-ventral coupling from each network z to network z + 1. (**D**) Bidirectional coupling from each network z to networks z − 1 and z + 1. (**E**) Left: coupling spread d(z) set to l(z) (dark) instead of a constant d (light). (**F**) Grid system with fewer networks h = 6 of smaller size n × n = 76 × 76. (**G**) Independent noise added to each neuron's firing rate at each timestep. (**H**) Coupling outputs randomly shifted for each neuron by one neuron in both x- and y-directions. (**I**) Spiking simulations with spikes generated by an independent Poisson process. Detailed methods for each system provided in Appendix 1.

DOI: https://doi.org/10.7554/eLife.46687.007

The following figure supplements are available for figure 4:

**Figure supplement 1.** Representative network activities and single neuron rate maps corresponding to *Figure 4A–C*.

DOI: https://doi.org/10.7554/eLife.46687.008

**Figure supplement 2.** Representative network activities and single neuron rate maps corresponding to *Figure 4D–F*.

DOI: https://doi.org/10.7554/eLife.46687.009

**Figure supplement 3.** Representative network activities and single neuron rate maps corresponding to *Figure 4G–I*.

DOI: https://doi.org/10.7554/eLife.46687.010

In our model, commensurate lattice relationships naturally lead to field-to-field firing rate variability in single neuron spatial rate maps (z = 8 in *Figure 2G*, for example), another experimentally observed feature of the grid system (*Ismakov et al., 2017*; *Dunn et al., 2017*; *Diehl et al., 2017*). At interfaces between two commensurate lattices, only a subset of population activity peaks in the grid of smaller scale overlap with, and thus receive excitation from, those in the grid of larger scale. The network with smaller grid scale will contain activity peaks of different magnitudes; this heterogeneity is then projected onto the spatial rate maps of its neurons.

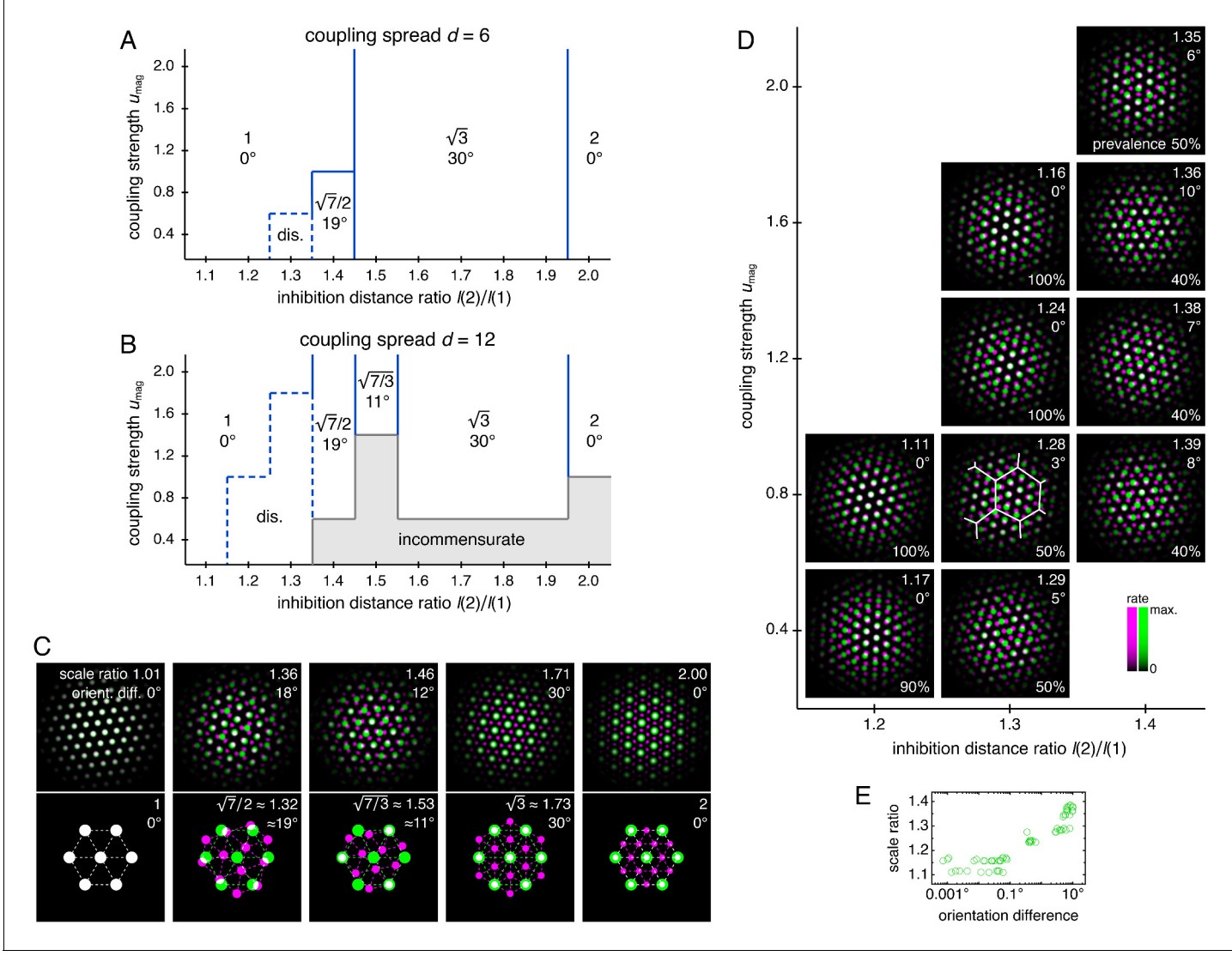

**Figure 5.** Diverse lattice relationships emerge over wide ranges in simulation parameters. In models with only two networks $z = 1$ and 2, we vary the coupling strength $u_{mag}$ and the ratio of inhibition distances $l(2)/l(1)$ for two different coupling spreads $d$. (**A, B**) Approximate phase diagrams based on 10 replicate simulations for each set of parameters, with the mean of $l(1)$ and $l(2)$ fixed to be 9. The most frequently occurring scale ratio and orientation difference are indicated for each region; coexistence between multiple lattice relationships may exist at drawn boundaries. (**A**) Phase diagram for small coupling spread $d = 6$. Solid lines separate four regions with different commensurate lattice relationships labeled by scale ratio and orientation difference, and dotted lines mark one region of discommensurate lattice relationships. (**B**) Phase diagram for large coupling spread $d = 12$. There are five different commensurate regions, a discommensurate region, as well as a region containing incommensurate lattices (gray). (**C**) Network activity overlays for representative observed (left) and idealized (right) commensurate relationships. Numbers at the top right of each image indicate network scale ratios $\lambda(2)/\lambda(1)$ and orientation differences $\theta(2) - \theta(1)$. Networks $z = 1$ and 2 in magenta and green, respectively, so white indicates activity in both networks. (**D**) Expanded region of **B** displaying discommensurate lattice statistics. For each set of parameters, a representative overlay for the most prevalent discommensurate lattice relationship is shown. The number in the lower right indicates the proportion of replicate simulations with scale ratio within 0.02 and orientation difference within 3° of the values shown at top right. In one overlay, discommensurations are outlined by white lines. (**E**) The discommensurate relationships described in **D** demonstrate positive correlation between scale ratio and the logarithm of orientation difference (Pearson's ρ = 0.91, $p \sim 10^{-26}$ ; Spearman's ρ = 0.92, p $\sim 10^{-27}$ ). Simulation details provided in Appendix 1.

DOI: https://doi.org/10.7554/eLife.46687.011

The following figure supplements are available for figure 5:

**Figure supplement 1.** Raw scale ratio and orientation difference data used to produce *Figure 5A*.

DOI: https://doi.org/10.7554/eLife.46687.012

**Figure supplement 2.** Raw scale ratio and orientation difference data used to produce *Figure 5B*.

DOI: https://doi.org/10.7554/eLife.46687.013

*Figure 5 continued on next page*

*Figure 5 continued*

**Figure supplement 3.** Commensurate and discommensurate relationships are robust against activity noise and coupling noise.
DOI: https://doi.org/10.7554/eLife.46687.014

## Excitation-inhibition balance sets lattice geometry

Adjusting the balance between excitatory coupling and a changing inhibition distance produces other commensurate lattice relationships, each of which enforces a certain scale ratio and orientation difference. To explore this competition systematically, we use a smaller coupled model with just two networks, $z = 1$ and $2$, and vary three parameters: the coupling spread $d$, the coupling strength $u_{mag}$, and the ratio of inhibition distances between the two networks $l(2)/l(1)$ (Appendix 1). For each set of parameters, we measure network scale ratios and orientation differences produced by multiple replicate simulations (*Figure 5—figure supplement 1* and *Figure 5—figure supplement 2*). We find that as the excitation-inhibition balance is varied by changing $u_{mag}$ and $l(2)/l(1)$, a number of discretely different relationships appear, which can be summarized in 'phase diagrams' (*Figure 5A, B*).

In many regions of the phase diagrams, these lattice relationships are commensurate, each with a characteristic scale ratio and orientation difference (*Figure 5C*). When parameters are chosen near a boundary between two regions, replicate simulations may adopt either lattice relationship or occasionally be trapped in other metastable relationships due to variations in random initial conditions (*Figure 5—figure supplement 2*). At larger $u_{mag}$ in both phase diagrams, there are fewer regions as $l(2)/l(1)$ varies because a higher excitatory coupling strength provides more rigidity against gradients in inhibition distance (*Figure 5A,B*). However, a larger coupling spread $d$ would cause network $z = 2$ to excite a broader set of neurons in network $z = 1$, softening the rigidity imposed by coupling and producing a wider variety of lattices in *Figure 5B* than *Figure 5A*. Also in *Figure 5B*, when excitation is weak and approaching the uncoupled limit, there is a noticeable region dominated by *incommensurate* lattices, in which the two grids lack consistent registry or relative orientation, and grid scale is largely determined by inhibition distance (*Figure 5—figure supplement 2*).

*Figure 5B* also contains a larger region of *discommensurate* lattices (although strictly speaking, in condensed matter physics, they would be termed commensurate lattices with discommensurations; *Chaikin and Lubensky, 1995*). Discommensurate networks have closely overlapping activities in certain areas that are separated by a mesh of regions lacking overlap called discommensurations (*Figure 5D*). They exhibit ranges of scale ratios 1.1–1.4 and orientation differences 0˚–10˚ that ultimately arise from a single source: the density of discommensurations, whose properties can also be explained through excitation-inhibition competition. Stronger coupling drives more activity overlap, which favors sparser discommensurations and lowers the scale ratio and orientation difference. However, a larger inhibition distance ratio drives the two networks to differ more in grid scale, which favors denser discommensurations. To better accommodate the discommensurations, grids rotate slightly as observed previously in a crystal system (*Wilson, 1990*). *Figure 5E* confirms that scale ratios and orientation differences vary together as the discommensuration density changes.

Thus, by changing the balance between excitation and inhibition, a two-network model yields geometric lattice relationships with various scale ratios and corresponding orientation differences. All the commensurate relationships (*Figure 5C*) and almost the entire range of discommensurate relationships (*Figure 5D*) have scale ratios that fall in the range of experimental measurements, which is roughly 1.2–2.0 (*Stensola et al., 2012*; *Barry et al., 2007*; *Krupic et al., 2015*). The scale ratios and orientation differences in both the commensurate and discommensurate cases are robust against activity noise and coupling noise (*Figure 5—figure supplement 3*).

## Discommensurate lattices produce distinct modular geometries but with more variation

As mentioned above, discommensurate lattices have a range of allowed geometries (*Figure 5D,E*), but they can still produce modules in a full 12-network grid system with a preferred scale ratio and orientation difference. However, these values do not cluster as strongly as they do for a commensurate relationship, which is geometrically precise.

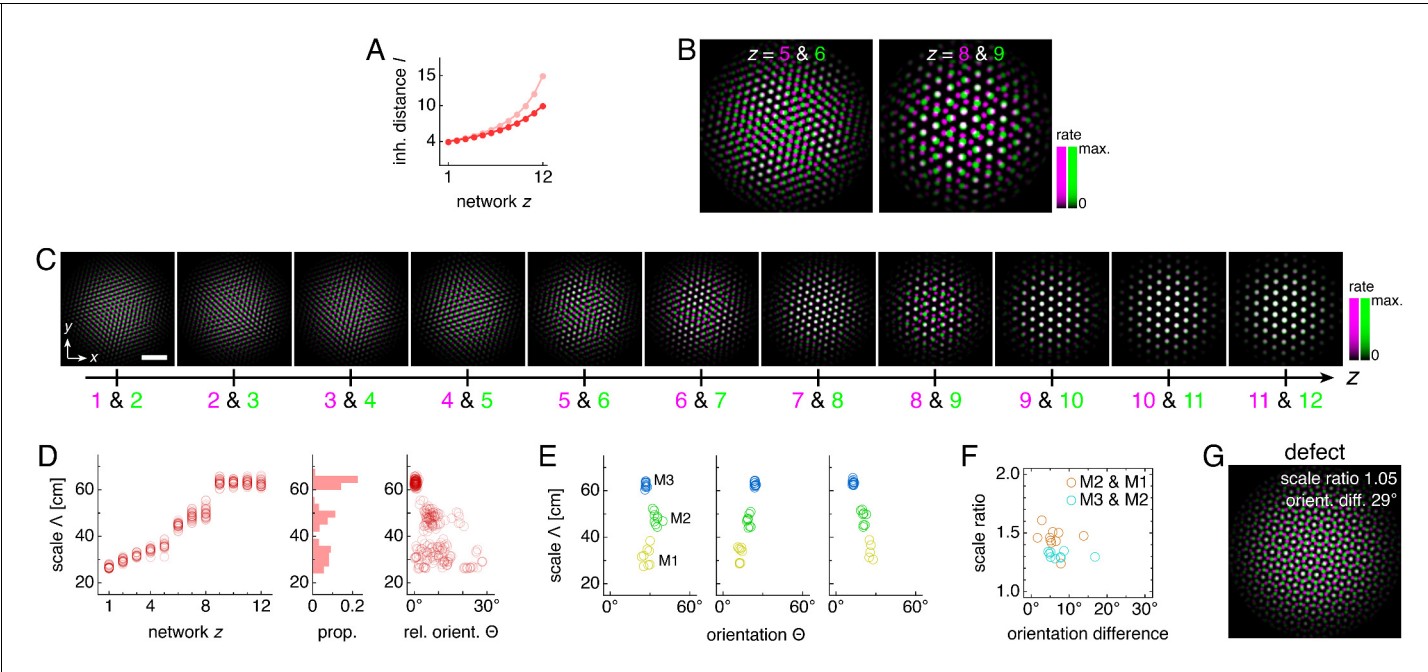

**Figure 6.** Discommensurate lattice relationships can produce realistic modules. (**A**) We use a shallower inhibition distance profile $l(z)$ (dark) compared to *Figure 1C* (light). (**B**) Large activity overlays from a representative simulation that emphasize discommensurate lattice relationships. (**C**) All activity overlays from the representative simulation in **B** between adjacent networks $z$ in magenta and green, so white indicates activity in both networks. Scale bar, 50 neurons. (**D–F**) Data from 10 replicate simulations. (**D**) Left: spatial grid scales $\Lambda(z)$. For each network, there are up to 30 red circles corresponding to three neurons recorded during each simulation. Middle: histogram for $\Lambda$ collected across all networks. Right: spatial orientations $\Theta$ relative to the grid cell in the same simulation with largest scale. (**E**) Clustering of spatial scales and orientations for three representative simulations. Due to sixfold lattice symmetry, orientation is a periodic variable modulo 60°. Different colors indicate separate modules. (**F**) Spatial scale ratios and orientation differences between adjacent modules. (**G**) Representative activity overlay demonstrating defect with low activity overlap. Maximum inhibition distance $l_{max} = 10$, coupling spread $d = 12$. We use larger network size $n \times n = 230 \times 230$ to allow for discommensurate relationships whose periodicities span longer distances on the neural sheets. Other parameter values are in *Table 1*.

DOI: https://doi.org/10.7554/eLife.46687.015

The following figure supplements are available for figure 6:

**Figure supplement 1.** Representative network activities and single neuron rate maps; module clustering for all replicate simulations.
DOI: https://doi.org/10.7554/eLife.46687.016
**Figure supplement 2.** Sample comparison of field-to-field firing rate variability between an experimental recording and our model.
DOI: https://doi.org/10.7554/eLife.46687.017

The phase diagrams of *Figure 5* provide guidance for modifying a 12-network system that exhibits a $[\sqrt{3}, 30°]$ relationship to produce discommensurate relationships instead. We make the inhibition distance profile $l(z)$ shallower (*Figure 6A*) and increase the coupling spread $d$ by 50%. Network activity overlays of these new simulations reveal grids obeying discommensurate relationships (*Figure 6B,C*), which are projected onto single neuron spatial rate maps through faithful path-integration (*Figure 6—figure supplement 1A*). Across replicate simulations with identical parameter values but different random initial firing rates, the discommensurate system demonstrates greater variation in scale and orientation (*Figure 6D*) than the commensurate system of *Figure 3* does. Nevertheless, analysis of each replicate simulation reveals clustering with well-defined modules (*Figure 6E* and *Figure 6—figure supplement 1B*). These modules have scale ratio $1.39 \pm 0.10$ and orientation difference $6.7 \pm 3.5°$ (mean ± s.d.; *Figure 6F*). The preferred scale ratio agrees well with the mean value observed experimentally in *Stensola et al. (2012)*.

Conceptually, we can interpret the greater spread of scales and orientations in terms of coupling rigidity. Excitatory coupling, especially when the spread is larger, provides enough rigidity in the discommensurate system to cluster scale ratios and orientation differences but not enough to prevent variations in these values. The degree of variability observed in *Figure 6D,E* appears consistent with

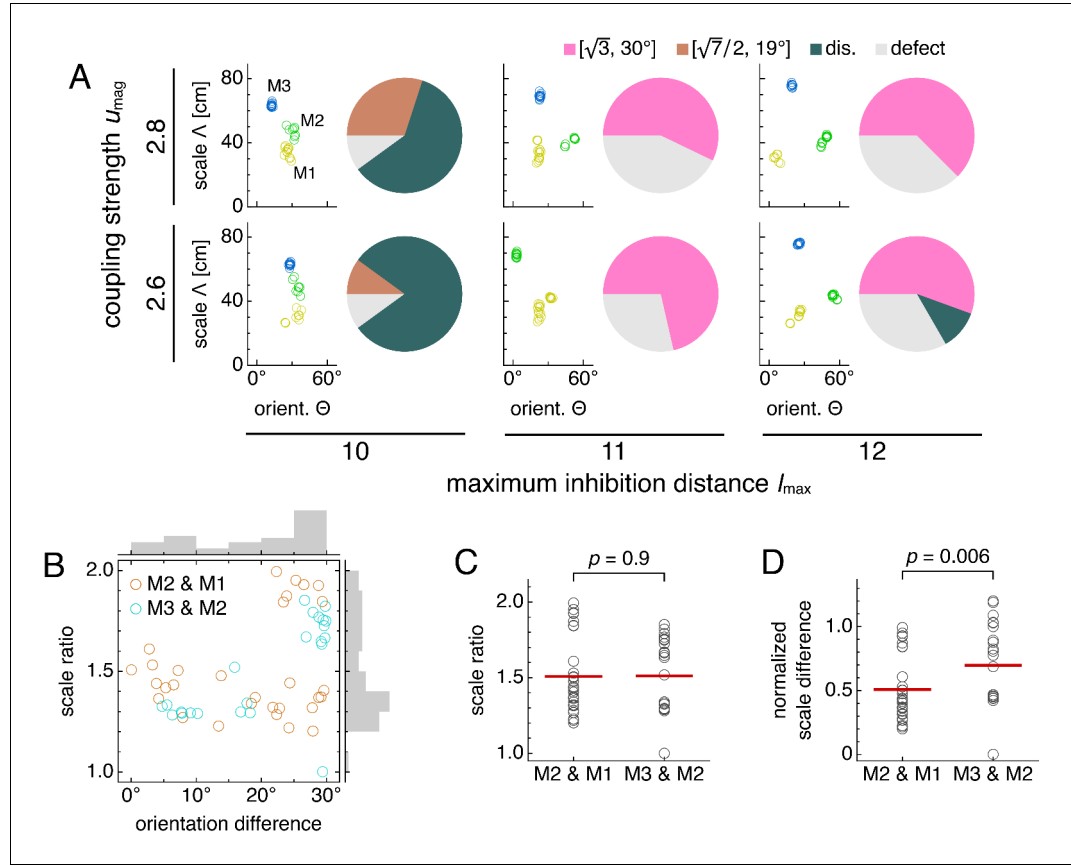

**Figure 7.** Simulations spanning different parameters contain diversity in lattice relationships, but average scale ratios are still constant between module pairs. Data from five replicate simulations for each set of parameters, encompassing 51 total module pairs. (A) Clustering of spatial scales and orientations for one representative simulation (left) and lattice relationship distribution across all pairs of adjacent modules (right) for each set of parameters. (B) Spatial scale ratios and orientation differences between adjacent modules with respective histograms to the right and above. Scale ratios and orientation differences exhibit positive rank correlation (Spearman's $\rho$ = 0.44, $p$ = 0.001). (C) Spatial scale ratios. Means indicated by lines. Medians compared through the Mann-Whitney $U$ test with reported $p$-value. (D) Spatial scale differences normalized by the scale of the first module (M1) in each simulation. Same interpretation of lines and $p$-value as in C. The $u_{mag}$ = 2.6 and $l_{max}$ = 10 data are taken from simulations in *Figure 5*. Some simulations produced only two modules M1 and M2; one simulation produced four modules, and M4 was excluded from further analysis. Coupling spread $d$ = 12 and network size $n \times n$ = 230 × 230. Other parameter values are in *Table 1*.

DOI: https://doi.org/10.7554/eLife.46687.018

The following figure supplements are available for figure 7:

**Figure supplement 1.** Module clustering for all simulations.

DOI: https://doi.org/10.7554/eLife.46687.020

**Figure supplement 2.** Lattice relationships that may underlie scale ratios and orientation differences for sample experimental recordings.

DOI: https://doi.org/10.7554/eLife.46687.019

experimental measurements, which also demonstrate spread (*Stensola et al., 2012*; *Barry et al., 2007*).

A few module pairs in *Figure 6F* exhibit a large orientation difference >10˚. This is not expected from a discommensurate relationship, and indeed, inspecting the network activities reveals adjacent networks trapped in a relationship with low activity overlap and large orientation difference (*Figure 6G*). In the context of a grid system that otherwise obeys commensurate or discommensurate geometries containing more overlap, we call this less common relationship a 'defect.' We distinguish between these relationships and the incommensurate lattices discussed above, which also

have low activity overlap. Defects arise when the excitatory coupling is strong, and incommensurate lattices arise when this coupling is weak. Also, defects have smaller scale ratios <1.1 and larger orientation differences >10°, whereas incommensurate lattices have larger scale ratios >1.3 and any orientation difference (*Figure 5B* and *Figure 5—figure supplement 2*).

Thus, networks governed by discommensurate relationships also cluster into modules with a preferred scale ratio and orientation difference within the experimental range (*Stensola et al., 2012*; *Krupic et al., 2015*). Due to lower coupling rigidity compared to commensurate grid systems, they exhibit increased variability and occasional defects across replicate simulations.

As in the commensurate case, discommensurate lattice relationships also create field-to-field firing rate variability in single neuron spatial rate maps. At interfaces between two discommensurate lattices, population activity peaks lack overlap at discommensurations and exhibit overlap in between them. Thus, only a subset of peaks in the grid of smaller scale receive excitation from the grid of larger scale; those located at discommensurations do not. As activity patterns translate on the neural sheets during path-integration, a grid cell in the network with smaller scale will have lower firing rate when a discommensuration moves through it, leading to firing rate variability (see *Figure 6—figure supplement 2* for an example).

## A diversity of lattice geometries maintains constant-on-average scale ratios

So far, each set of 12-network simulations contained replicates with identical parameter values and exhibited a single dominant lattice relationship. We now present results with different parameter values to imitate biological network variability across animals. This procedure leads to modules with different commensurate and discommensurate relationships (*Figure 7A* and *Figure 7—figure supplement 1*). There is no longer a single preferred scale ratio or orientation difference (*Figure 7B*), but patterns emerge due to the predominance of discommensurate and commensurate relationships. Recall from *Figure 6F* that discommensurate module pairs exhibit scale ratios ≈1.4 and orientation differences ≈7°. Combined with $[\sqrt{3} \approx 1.7, 30°]$ module pairs, we find a bimodal distribution of orientation differences around 7° and 30°, consistent with experimental data (*Krupic et al., 2015*), and positive correlation between scale ratio and orientation difference. Modules with low scale ratio but high orientation difference decrease this correlation; they arise from defects (*Figure 6G*). *Figure 7—figure supplement 2* illustrates how modules observed experimentally may be governed by a variety of lattice relationships.

Scale ratios across the assorted simulations span a range of values, but their averages are constant across module pairs. That is, the median scale ratio does not change between the pair of modules with smaller scales and the larger pair (*Figure 7C*). Similarly, mean values are respectively $1.52 \pm 0.05$ and $1.53 \pm 0.05$ (mean ± s.e.m.) for module pairs M2 and M1 and M3 and M2. Combining data from both module pairs gives scale ratio $1.52 \pm 0.03$ (mean ± s.e.m.), which agrees well with the mean value of 1.56 from *Krupic et al. (2015)*. *Stensola et al. (2012)* reports a slightly smaller mean value of $1.42 \pm 0.17$ (mean ± s.d.; re-analyzed by *Wei et al., 2015*), but its broad distribution of scale ratios overlaps considerably with ours. Moreover, we find that the normalized scale *difference* does change its median across module pairs (*Figure 7D*). This result that scale ratios are constant on average but scale differences are not matches experiment (*Stensola et al., 2012*).

Thus, although our model can produce modules with fixed scale ratios, allowing for a range of network parameters also produces modules with a range of scale ratios. Nevertheless, the scale ratio averaged over these parameters is still constant across module pairs, a key feature of the grid system that holds even if scales are not governed by a universal ratio (*Stensola et al., 2012*).

## Testing for coupling: a mock lesion experiment

Excitatory coupling locks networks into scales and orientations imposed by more ventral networks. Disrupting the coupling frees networks from this rigidity, which can change scales and orientations far from the disruption. We demonstrate this effect by inactivating one network $z = 7$ midway through the simulation (*Figure 8A*). This corresponds experimentally to disrupting excitatory connections at one location along the dorsoventral MEC axis.

After the lesion, grid cells ventral to the lesion location ($z \geq 8$) are unaffected, but those dorsal to the lesion location ($z \leq 6$) change scale and orientation and form a single module (*Figure 8B–D*).

Network $z = 6$ is no longer constrained by larger grids of more ventral networks, so its scale decreases. The coupling that remains from $z = 6$ to $1$ then rigidly propagates the new grid down to network $z = 1$. This post-lesion module M1 has larger scale and 30° orientation difference compared to the pre-lesion M1; these changes also appear as corresponding changes in the scale ratio and orientation difference between modules M2 and M1 (*Figure 8E*).

Immediate changes in grid scale and/or orientation observed at one location along the longitudinal MEC axis due to a lesion at another location would strongly support the presence of the excitatory coupling predicted by our model. Moreover, the anatomical distribution of the changes would indicate the directionality of coupling; those in grid cells dorsal to the lesion would indicate ventral-to-dorsal coupling and those ventral to the lesion would indicate dorsal-to-ventral coupling.

We have also considered the consequences of certain incomplete lesions. A regional lesion, in which a corner of the lesioned network $z = 7$ is preserved, causes each more dorsal network to contain regions with different scales (*Figure 8—figure supplement 1* and *Figure 8—video 1*). These differences are not large enough to create a new module close to the lesioned network ($z = 5$ and $6$), so scale ratios and orientations are not strongly affected. However, different regions of each network will independently transition to the smallest module farther away from the lesioned network ($z = 1$ to $4$). Thus, one network corresponding to a single location along the dorso-ventral MEC axis can contain grid cells belonging to two modules. Experimentally, grid modules do overlap in their anatomic extent along the MEC axis (*Stensola et al., 2012*); our model predicts that this overlap may be enhanced by a regional lesion. Note that some neurons also appear to show band-like spatial rate maps ($z = 4$ and $6$ in *Figure 8—figure supplement 1A*), whose experimental observation has been reported (*Krupic et al., 2012*) but disputed (*Navratilova et al., 2016*). We also performed a decimation-type lesion, in which one neuron of every $3 \times 3$ block is preserved in the lesioned network. This impedes the motion of the grid pattern on the neural sheet in more dorsal networks (*Figure 8—video 2*) and thus destroys single neuron grid responses in those networks (*Figure 8—figure supplement 1D*).

## Discussion

We propose that the hierarchy of grid modules in the MEC is self-organized by competition in attractor networks between excitation along the longitudinal MEC axis and lateral inhibition. We showed that such an architecture, with an inhibition distance that increases smoothly along the MEC axis, reproduces a central experimental finding: grid cells form modules with scales clustered around discrete values (*Stensola et al., 2012*; *Barry et al., 2007*; *Krupic et al., 2015*).

The distribution of scales across modules in our model quantitatively matches experiments. Different groups have reported mean scale ratios of 1.64 (6 module pairs), 1.42 (24 module pairs), and 1.56 (11 module pairs) (*Barry et al., 2007*; *Stensola et al., 2012*; *Krupic et al., 2015*). These data could be interpreted as an indication that the grid system has a preferred scale ratio roughly in range of 1.4–1.7. As we showed, our model naturally produces a hierarchy of modules with scale ratios in this range; its network parameters lead to both commensurate and discommensurate grids (*Figure 5*). On the other hand, the data on scale ratios between individual pairs of modules actually span a range of values in the different experiments: 1.6–1.9, 1.1–1.8, and 1.2–2.0 (*Barry et al., 2007*; *Stensola et al., 2012*; *Krupic et al., 2015*). This suggests that the underlying mechanism that produces grid modules must be capable of producing different scale ratios as its parameters vary. This is indeed the case for our model, in which variation of network parameters produces a realistic range of scale ratios (*Figure 7*). Despite variability across individual scale ratios, experiments strikingly reveal that the average scale ratio is the same from the smallest pair of modules to the largest pair, whereas the average scale *difference* changes across the hierarchy (*Stensola et al., 2012*). Our model robustly reproduces this observation (*Figure 7C,D*) because its fundamental mechanism of geometric coordination between grids enforces constant-on-average scale ratios even with variation in parameters among individual networks.

Our model requires that grid orientation be co-modular with scale, as observed in experiment (*Stensola et al., 2012*). Studies characterizing the statistics of orientation differences between modules are limited, but values seem to span the entire range 0˚–30˚, with some preference for values at the low and high ends of this range (*Krupic et al., 2015*). Our model can capture the entire range of orientation differences with discommensurate relationships favoring small differences and

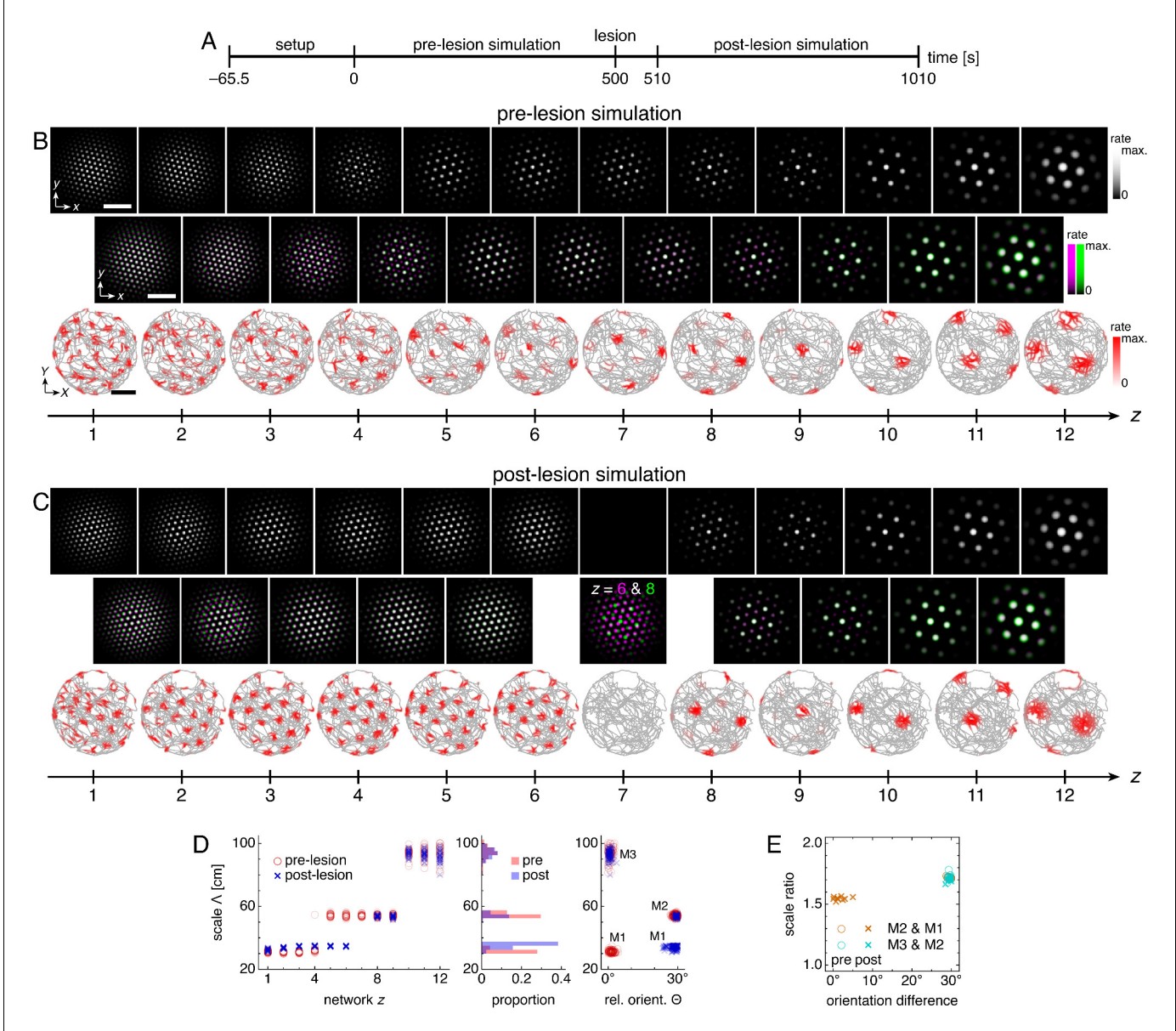

**Figure 8.** Lesioning a network changes grid scales and orientations of more dorsal networks. (**A**) Lesion protocol. The lesion inactivates network $z = 7$. (**B**) A representative simulation before the lesion. Top row: network activities at the end of the pre-lesion simulation. Second row: activity overlays between adjacent networks depicted in the top row. In each panel, the network with smaller (larger) $z$ is depicted in magenta (green), so white indicates activity in both networks. Third row: spatial rate map of a single neuron for each $z$ superimposed on the animal's trajectory. White scale bars, 50 neurons. Black scale bars, 50 cm. (**C**) Same as **B** but after the lesion. Spatial rate maps are recorded from the same neurons as in **B**. (**D, E**) Data from 10 replicate simulations. (**D**) Left: spatial grid scales $\Lambda(z)$ before and after the lesion. Middle: histogram for $\Lambda$ collected across all networks. Right: spatial orientations $\Theta$ relative to the grid cell in the same simulation with largest scale. (**E**) Spatial scale ratios and orientation differences between adjacent modules. Standard parameter values provided in *Table 1*.

DOI: https://doi.org/10.7554/eLife.46687.021

The following video and figure supplement are available for figure 8:

**Figure supplement 1.** The effects of incomplete lesions on grid cells in more dorsal networks.

DOI: https://doi.org/10.7554/eLife.46687.022

**Figure 8—video 1.** Last 100 s of the simulation displayed in *Figure 8—figure supplement 1A*.

DOI: https://doi.org/10.7554/eLife.46687.023

**Figure 8—video 2.** Last 100 s of the simulation displayed in *Figure 8—figure supplement 1D*.

DOI: https://doi.org/10.7554/eLife.46687.024

commensurate relationships favoring large differences (*Figure 5*). Overall, our model predicts a positive correlation between scale ratio and orientation difference (*Figure 5E* and *Figure 7B*), which can be tested experimentally. Existing datasets (*Stensola et al., 2012*; *Krupic et al., 2015*) have a confound—animals are tested in square and rectangular enclosures which have distinguishable orientations marked by the corners. Grid orientations can anchor to such features (*Stensola et al., 2015*), either through the integration of visual and external cues (*Raudies and Hasselmo, 2015*; *Savelli et al., 2017*), or through interaction with boundaries (*Bush and Burgess, 2014*; *Krupic et al., 2016*; *Giocomo, 2016*; *Evans et al., 2016*; *Hardcastle et al., 2017*; *Keinath et al., 2018*; *Ocko et al., 2018*). Experiments in circular or other non-rectangular environments may help disambiguate the effects of such anchoring. Our model also predicts that orientation differences between modules will be preserved between environments with different geometries since the differences are internally generated by the dynamics of the network. This effect has been observed (*Krupic et al., 2015*).

Our model produces consistent differences in firing rate from one grid field to another for some grid cells. This variability is structured because it arises at module interfaces from the selective excitation of some network activity peaks in the smaller-scale grid by the overlapping activity peaks of the larger-scale grid. Such an explanation for firing rate variability has been suggested by *Ismakov et al. (2017)*. Signatures of structured variability can be sought in experimental grid cell recordings (see *Figure 6—figure supplement 2* for an example). However, these signatures may be obscured by other sources of grid variability, such as proposed inputs from place cells (*Dunn et al., 2017*) and the observed modulation of grid fields by reward (*Butler et al., 2019*; *Boccara et al., 2019*), which may in turn be also related to hippocampal input.

Our model requires excitatory coupling between grid cells at different locations along the longitudinal MEC axis, either through direct excitation or disinhibition (*Fuchs et al., 2016*). Short-range excitatory connections between principal neurons in superficial MEC layers have been discovered recently through patch clamp experiments (*Fuchs et al., 2016*; *Winterer et al., 2017*). These neurons also make long-range projections to superficial layers of the contralateral MEC (*Varga et al., 2010*; *Fuchs et al., 2016*), where they connect to other principal cells (*Zutshi et al., 2018*). The validity of our model would be bolstered if similar connections were found between locations along the MEC that correspond to different grid modules.

The presence of excitatory coupling can also be tested indirectly. We predict that the destruction of grid cells, or inactivation of excitatory coupling (*Zutshi et al., 2018*), at a given location along the axis will change grid scales and/or orientations at other locations (*Figure 8*). The presence of noise correlations across modules, as previously investigated but not fully characterized (*Mathis et al., 2013*; *Tocker et al., 2015*), would suggest connections between modules. Such correlations, and perhaps even lattice relationships, could be observed via calcium imaging of the MEC (*Heys et al., 2014*; *Gu et al., 2018*). The effect of environmental manipulations on grid relationships has been suggested to demonstrate both independence (*Stensola et al., 2012*) and dependence (*Krupic et al., 2015*) across modules. However, (*Keinath et al., 2018*) showed that apparent deformations of grids after changes in environmental shape may result in part from learned interactions with boundaries, perhaps mediated by border cells. Thus, environmental deformation paradigms may not be ideal tests of our model due to confounding boundary effects (*Keinath et al., 2018*; *Ocko et al., 2018*).

Our predictions may be altered by synaptic plasticity, which we do not implement in our model. Spike-timing-dependent plasticity rules are capable of creating the recurrent inhibitory architecture required by continuous attractor models of a single grid module (*Widloski and Fiete, 2014*). As for our model with multiple modules, synaptic plasticity within the inhibitory connections may resolve the competition between excitation and inhibition by adjusting the inhibition distance in each network to the value favored by the rigidity of excitatory coupling. In that case, lesioning one network would not affect the grid scales of other networks, although changes in orientation differences may be observed over time due to attractor drift. Nevertheless, our proposed geometric mechanism could still govern the initial formation of modules with certain scale ratios before plasticity fully takes effect.

Since spatial grid scales are both proportional to inhibition distance $l$ and inversely proportional to velocity gain $\alpha$ (*Burak and Fiete, 2009* and Materials and methods), we also simulated excitatorily coupled networks with a depth-dependent velocity gain $\alpha(z)$ and a fixed inhibition distance $l$

**Table 1.** Main model parameters and their values unless otherwise noted.

| Parameter | Variable | Value |
|---|---|---|
| Number of networks | $h$ | 12 |
| Number of neurons per network | $n \times n$ | $160 \times 160$ |
| Neurons recorded per network | | 3 |
| Animal speed | $|\mathbf{V}|$ | 0–1 m/s |
| Diameter of enclosure | | 180 cm |
| Simulation time | | 500 s |
| Simulation timestep | $\Delta t$ | 1 ms |
| Neural relaxation time | $\tau$ | 10 ms |
| Broad input strength | $a_{mag}$ | 1 |
| Broad input falloff | $a_{fall}$ | 4 |
| Inhibition distance minimum | $l_{min}$ | 4 |
| Inhibition distance maximum | $l_{max}$ | 15 |
| Inhibition distance exponent | $l_{exp}$ | −1 |
| Inhibition strength | $w_{mag}$ | 2.4 |
| Subpopulation shift | $\xi$ | 1 |
| Coupling spread | $d$ | 8 |
| Coupling strength | $u_{mag}$ | 2.6 |
| Velocity gain | $\alpha$ | 0.3 s/m |

DOI: https://doi.org/10.7554/eLife.46687.006

(Appendix 2). In contrast to simulations in one dimension (J Widloski and I Fiete, personal communication, October 2017), while we observed module self-organization, the system gave inconsistent results among replicate simulations and lacked fixed scale ratios. Moreover, recent calcium imaging experiments suggest that activity on the MEC is arranged a deformed triangular lattice (*Gu et al., 2018*), as predicted by the continuous attractor model (*Burak and Fiete, 2009*), and that regions with activity separated by larger anatomic distances contain grid cells of larger spatial scale. These observations support a changing inhibition distance over a changing velocity gain as a mechanism for producing different grid scales, under the assumption that anatomic and network distances correspond to each other.

Our results differ from previous work on mechanisms for forming grid modules. Grossberg and Pilly hypothesize that grid cells arise from stripe cells in parasubiculum, and that discreteness in the spatial period of stripe cells leads to modularity of grid cells (*Grossberg and Pilly, 2012*). However, stripe cells have only been observed once (*Krupic et al., 2012*; *Navratilova et al., 2016*), and the origin of discrete periods with constant-on-average ratios in stripe cells would then need to be addressed. Urdapilleta, Si, and Treves propose a model in which discrete modules self-organize from smooth gradients in parameters in a model where grid formation is driven by firing rate adaptation in single cells (*Urdapilleta et al., 2017*). They also utilize excitatory coupling among grid cells along the longitudinal MEC axis. However, this model does not have a mechanism to dynamically enforce the average constancy of grid scale ratios, which is a feature of the grid system (*Stensola et al., 2012*). Furthermore, it produces modules with orientation differences near zero and does not demonstrate values near 30° (*Krupic et al., 2015*). Our model naturally produces constant-on-average scale ratios and allows for a wide range of orientation differences. Moreover, over the past few years, multiple reports have provided independent experimental support for the importance of recurrent connections among grid cells (*Couey et al., 2013*; *Dunn et al., 2015*; *Fuchs et al., 2016*; *Zutshi et al., 2018*) and for the continuous attractor model in particular (*Yoon et al., 2013*; *Heys et al., 2014*; *Gu et al., 2018*). Our work establishes that continuous attractor networks can produce a discrete hierarchy of modules with a constant-on-average scale ratio.

The competition generated between excitatory and inhibitory connections bears a strong resemblance to the Frenkel-Kontorova model of condensed matter physics, in which a periodic potential of one scale acts on particles that prefer to form a lattice of a different, competing scale (**Kontorova and Frenkel, 1938**). This model has a rich literature with many deep theoretical results, including the calculation of complicated phase diagrams involving 'devil's staircases' (**Bak, 1982**; **Chaikin and Lubensky, 1995**) which mirror those of our model (**Figure 5**). Under certain conditions, our model produces networks with quasicrystalline approximant grids that are driven by networks with standard triangular grids at other scales (Appendix 3). Quasicrystalline order lacks periodicity, but contains more nuanced positional order (**Levine and Steinhardt, 1986**). This phenomenon wherein quasicrystalline structure is driven by crystalline order in a coupled system was recently observed for the first time in thin-film materials that contain Frenkel-Kontorova-like interactions (**Förster et al., 2013**; **Förster et al., 2016**; **Paßens et al., 2017**).

Commensurate and discommensurate lattice relationships are a robust and versatile mechanism for self-organizing a grid system whose scale ratios are constant or constant on average across a hierarchy of modules. We demonstrated this mechanism in a basic extension of the continuous attractor model with excitatory connections between networks. This model is amenable to extensions that capture other features of the grid system, such as fully spiking dynamics, learning of synaptic weights (**Widloski and Fiete, 2014**), the union of our separate networks into a single network spanning the entire MEC, and the addition of border cell inputs or recurrent coupling between modules to correct path-integration errors or react to environmental deformations (**Hardcastle et al., 2015**; **Keinath et al., 2018**; **Ocko et al., 2018**; **Pollock et al., 2017**; **Mosheiff and Burak, 2019**).

## Materials and methods

### Model setup and dynamics

We implemented the Burak-Fiete model (**Burak and Fiete, 2009**) as follows (**Source code 1**). Networks $z = 1, \ldots, h$ each contain a 2D sheet of neurons with indices $\mathbf{r} = (x, y)$, where $x = 1, \ldots, n$ and $y = 1, \ldots, n$. Neurons receive broad excitatory input $a(\mathbf{r})$ from the hippocampus, and, to prevent edge effects, those toward the center of the networks receive more excitation than those toward the edges. Each neuron also inhibits others that lie around a length scale of $l(z)$ neurons away in the same network $z$. Moreover, every neuron belongs to one of four subpopulations that evenly tile the neural sheet. Each subpopulation is associated with both a preferred direction $\hat{\mathbf{e}}$ along one of the network axes $\pm\hat{\mathbf{x}}$ or $\pm\hat{\mathbf{y}}$ and a corresponding preferred direction $\hat{\mathbf{E}}$ along an axis $\pm\hat{\mathbf{X}}$ or $\pm\hat{\mathbf{Y}}$ in its spatial environment. A neuron at position $\mathbf{r}$ in network $z$ has its inhibitory outputs $w(\mathbf{r}; z)$ shifted slightly by $\xi$ neurons in the $\hat{\mathbf{e}}(\mathbf{r})$ direction and its broad excitation modulated by a small amount proportional to $\hat{\mathbf{E}}(\mathbf{r}) \cdot \mathbf{V}$, where $\mathbf{V}$ is the spatial velocity of the animal. Note that lowercase letters refer to attractor networks at each depth $z$ in which distances have units of neurons, and uppercase letters refer to the animal's spatial environment in which distances have physical units, such as centimeters.

In addition to these established features (**Burak and Fiete, 2009**), we introduce excitatory connections $u(\mathbf{r})$ from every neuron $\mathbf{r}$ in network $z$ to neurons located within a spread $d$ of the same $\mathbf{r}$ but in the preceding network with depth $z - 1$. $u(\mathbf{r})$ is constant for all networks. These components lead to the following dynamical equation for the dimensionless neural firing rates $s(\mathbf{r}, z, t)$:

$$\tau \frac{s(\mathbf{r}, z, t + \Delta t) - s(\mathbf{r}, z, t)}{\Delta t} + s(\mathbf{r}, z, t)$$

$$= \left\{ \sum_{\mathbf{r}'} w(\mathbf{r} - \mathbf{r}' + \xi\hat{\mathbf{e}}(\mathbf{r}'); z) s(\mathbf{r}', z, t) + \sum_{\mathbf{r}'} u(\mathbf{r} - \mathbf{r}') s(\mathbf{r}', z + 1, t) + a(\mathbf{r}) \left[ 1 + \alpha\hat{\mathbf{E}}(\mathbf{r}) \cdot \mathbf{V}(t) \right] \right\}_+. \tag{1}$$

Inputs to each neuron are rectified by $\{c\}_+ = 0$ for $c < 0$, $c$ for $c \geq 0$. $\Delta t$ is the simulation time increment, $\tau$ is the neural relaxation time, and $\alpha$ is the velocity gain that describes how much the animal's velocity $\mathbf{V}$ modulates the broad inputs $a(\mathbf{r})$. Note that $s$ can be treated as a dimensionless variable because **Equation 1** is invariant to scaling of $s$ and $a$ by the same factor.

We use velocities $\mathbf{V}(t)$ corresponding to a real rat trajectory (**Hafting et al., 2005**; **Burak and Fiete, 2009**). Details are provided in Appendix 1.

## Inhibitory and excitatory connections

The broad excitatory input is

$$a(\mathbf{r}) = \begin{cases} a_{\text{mag}} e^{-a_{\text{fall}} r_{\text{scaled}}^2} & r_{\text{scaled}} < 1 \\ 0 & r_{\text{scaled}} \geq 1, \end{cases}$$

(2)

where $r_{scaled} = \sqrt{\left(x - \frac{n+1}{2}\right)^2 + \left(y - \frac{n+1}{2}\right)^2} / \frac{n}{2}$ is a scaled radial distance for the neuron at $\mathbf{r} = (x, y)$, $a_{mag}$ is the magnitude of the input, and $a_{fall}$ is a falloff parameter. The inhibition distance for network $z$ is

$$l(z) = \left[ l_{min}^{l_{exp}} + \left( l_{max}^{l_{exp}} - l_{min}^{l_{exp}} \right) \frac{z - 1}{h - 1} \right]^{1/l_{exp}},$$

(3)

which ranges from $l_{min} = l(1)$ to $l_{max} = l(h)$ with concavity tuned by $l_{exp}$. More negative values of $l_{exp}$ lead to greater concavity; for $l_{exp} = 0$, we use the limiting expression $l(z) = l_{min}^{(h-z)/(h-1)} l_{max}^{(z-1)/(h-1)}$. The recurrent inhibition profile for network $z$ is

$$w(\mathbf{r}; z) = \begin{cases} -\frac{w_{mag}}{l(z)^2} \frac{1 - \cos[\pi r / l(z)]}{2} & r < 2l(z) \\ 0 & r \geq 2l(z), \end{cases}$$

(4)

where $w_{mag}$ is the magnitude of inhibition. We scale this magnitude by $l(z)^{-2}$ to make the integrated inhibition constant across $z$. The excitatory coupling is

$$u(\mathbf{r}) = \begin{cases} \frac{u_{mag}}{d^2} \frac{1 + \cos[\pi r / d]}{2} & r < d \\ 0 & r \geq d, \end{cases}$$

(5)

where $u_{mag}$ and $d$ are the magnitude and spread of coupling, respectively. In analogy to $w_{mag}$, we scale $u_{mag}$ by $d^{-2}$.

## Overview of data analysis techniques

To determine spatial grid scales, orientations, and gridness, we consider an annular region of the spatial autocorrelation map that contains the six peaks closest to the origin. Grid scale is the radius with highest value, averaging over angles. Grid orientation and gridness are determined by first averaging over radial distance and analyzing the sixth component of the Fourier series with respect to angle (*Weber and Sprekeler, 2019*). The power of this component divided by the total Fourier power measures 'gridness' and its complex phase measures the orientation. Grid cells are subject to a gridness cutoff of 0.6. For each replicate simulation, we cluster its grid cells with respect to scale and orientation using a $k$-means procedure with $k$ determined by kernel smoothed densities (*Stensola et al., 2012*). See Appendix 1 for full details.

## Acknowledgements

We are grateful to Xue-Xin Wei, Tom Lubensky, Ila Fiete, John Widloski, and Zengyi Li for their thoughtful ideas and suggestions. We thank Hanne Stensola and Julija Krupic for sharing raw experimental data. We are also grateful to the Kavli Institute for the Physics and Mathematics of the Universe for hospitality provided to VB.

## Additional information

### Funding

| Funder | Grant reference number | Author |
| --- | --- | --- |
| Honda Research Institute | Embodied, efficient, geometry-driven curiosity | Vijay Balasubramanian |
| National Science Foundation | PHY-1734030 | Vijay Balasubramanian |

| Adolph C. and Mary Sprague Miller Institute for Basic Research in Science, University of California Berkeley | Postdoctoral fellowship | Louis Kang |
| National Institutes of Health | Medical Scientist Training Program | Louis Kang |

The funders had no role in study design, data collection and interpretation, or the decision to submit the work for publication.

### Author contributions

Louis Kang, Conceptualization, Software, Investigation, Visualization, Methodology, Writing—original draft, Writing—review and editing; Vijay Balasubramanian, Conceptualization, Resources, Supervision, Funding acquisition, Visualization, Methodology, Writing—original draft, Writing—review and editing

### Author ORCIDs

Louis Kang  https://orcid.org/0000-0002-5702-2740
Vijay Balasubramanian  https://orcid.org/0000-0002-6497-3819

### Decision letter and Author response

Decision letter https://doi.org/10.7554/eLife.46687.035
Author response https://doi.org/10.7554/eLife.46687.036

## Additional files

### Supplementary files

• Source code 1. Source code for the main simulations written in C.
DOI: https://doi.org/10.7554/eLife.46687.025

• Transparent reporting form
DOI: https://doi.org/10.7554/eLife.46687.026

### Data availability

We have included the source code for our main simulation as a supporting file.

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

## Appendix 1

DOI: https://doi.org/10.7554/eLife.46687.027

### Additional methods

#### Simulation setup

#### Standard model

To distribute neural subpopulations evenly, we assign each position in a $2 \times 2$ block of neurons to a different subpopulation and tile each network with these blocks. In other words, for a network of size $n \times n$, the preferred network directions are $\hat{e}(2i-1, 2j-1) = -\hat{x}$, $\hat{e}(2i-1, 2j) = \hat{y}$, $\hat{e}(2i, 2j-1) = -\hat{y}$, and $\hat{e}(2i, 2j) = \hat{x}$ for block indices $i, j = 1, \ldots, n/2$. The preferred spatial directions take corresponding values $\hat{E}(2i-1, 2j-1) = -\hat{X}$, $\hat{E}(2i-1, 2j) = \hat{Y}$, $\hat{E}(2i, 2j-1) = -\hat{Y}$, and $\hat{E}(2i, 2j) = \hat{X}$.

We initialize each neuron with a uniformly-distributed random firing rate between 0 and 0.001 (arbitrary units). We evolve 500 timesteps without velocity input to generate grid-like activity. Next, we anneal grid defects. For each velocity angle $\pi/2 - \pi/5$, $2\pi/5$, and $\pi/4$, we evolve 5000–10000 timesteps with constant speed $0.5\,\mathrm{m/s}$. We then evolve 50000 timesteps with velocity data from a real rat trajectory within a circular enclosure (*Hafting et al., 2005*; *Burak and Fiete, 2009*). The main simulation phase ensues with continuation of velocity input from the trajectory. For each network $z$, we randomly choose three neurons within a distance of $0.15n$ from the network center. Throughout the main phase, we tabulate their mean firing rates as a function of rat spatial position.

#### Modified models in *Figure 4A–H*

The various models depicted in *Figure 4* differ from the standard model with standard parameters in *Table 1* in the following ways.

*Figure 4A* $l_{exp} = 0$.

*Figure 4B* $l_{exp} = -2$.

*Figure 4C* Dorsal-to-ventral coupling from each network $z$ to network $z + 1$, with $u_{mag} = 0.8$ and $d = 1$.

*Figure 4D* Bidirectional coupling from each network $z$ to networks $z - 1$ and $z + 1$, with $u_{mag} = 0.4$ and $d = 1$.

*Figure 4E* Coupling spread $d(z)$ set to $l(z)$.

*Figure 4F* Fewer networks $h = 6$ of size $n \times n = 76 \times 76$. $l_{min} = 2.4$, $l_{max} = 9$, $w_{mag} = 2.0$, $d = 2$, $u_{mag} = 1.2$, $\alpha = 1.8$, $a_{fall} = 3$. Main simulation run for 300000 timesteps.

*Figure 4G* Independent Gaussian noise with mean 0 and standard deviation 0.3 added to neural inputs; that is, this noise term is introduced inside the braces of *Equation 1* in Materials and methods. $w_{mag} = 1.8$, $d = 4$, $u_{mag} = 1.2$. Main simulation run for 300000 timesteps.

*Figure 4H* The excitatory outputs for neuron $(x, y)$ in network $z$ are centered at $(x \pm 1, y \pm 1)$ in network $z = 1$, with signs randomly chosen for each neuron. $d = 4$, $u_{mag} = 1.6$. Main simulation run for 300000 timesteps.

#### Spiking model in *Figure 4I*

We follow *Burak and Fiete (2009)* and simulate stochastic spiking with sub-Poisson statistics. Firing rates (*Equation 1* in Materials and methods) are replaced by synaptic activations $s(\mathbf{r}, z, t)$ that evolve as

$$\tau \frac{s(\mathbf{r}, z, t + \Delta t) - s(\mathbf{r}, z, t)}{\Delta t} + s(\mathbf{r}, z, t) = \frac{p(\mathbf{r}, z, t)}{\Delta t}, \tag{6}$$

where $p(\mathbf{r}, z, t) = 1$ if neuron $\mathbf{r} = (x, y)$ in network $z$ spikes at time $t$ or $p(\mathbf{r}, z, t) = 0$ if it does not. One can recover a firing rate interpretation in a deterministic limit by choosing $\Delta t$ to be the

fixed time between regular spikes. In that case, $s(\mathbf{r}, z, t + \Delta t) = s(\mathbf{r}, z, t)$ and $p(\mathbf{r}, z, t) = 1$, so $s(\mathbf{r}, z, t) = 1/\Delta t$, which is the firing rate.

The rate parameter of the spiking process is governed by the total neural input

$$s_{in}(\mathbf{r}, z, t) = \left\{ \sum_{\mathbf{r}'} w(\mathbf{r} - \mathbf{r}' + \xi \hat{\mathbf{e}}(\mathbf{r}'); z) s(\mathbf{r}', z, t) + \sum_{\mathbf{r}'} u(\mathbf{r} - \mathbf{r}') s(\mathbf{r}', z + 1, t) + a(\mathbf{r}) \left[ 1 + \alpha \hat{\mathbf{E}}(\mathbf{r}) \cdot \mathbf{V}(t) \right] \right\}_+ \quad (7)$$

(c.f. *Equation 1* in Materials and methods). To generate a sub-Poisson process whose interspike intervals exhibit coefficient of variation $CV = 1/\sqrt{m}$, for each neuron, we sample $m$ times from a Bernoulli distribution with probability $s_{in}(\mathbf{r}, z, t)\Delta t$. We take every $m$th 1 to be a spike and discard all other results. Note that unlike the rate model, $s$ can no longer be treated as a dimensionless variable.

We use $w_{mag} = 2.0$, $d = 4$, $u_{mag} = 1.0$, $a_{mag} = 0.6$, $a_{fall} = 3$. We use a 10-fold finer simulation timestep $\Delta t = 0.1\,\text{ms}$ and run each phase of the simulation setup for 10 times more timesteps. We run the main simulation for 2000000 timesteps.

## Two-network model for phase diagrams in *Figure 5*

To generate data for the phase diagrams in *Figure 5*, we set up our simulations in a similar way, with the following differences. We use only two network depths. We use slightly larger velocity gain $\alpha = 0.4\,\text{s/m}$ to produce grids of smaller spatial scale since a greater number of activity peaks allows for better measurement of grid scale. After initializing the system and performing initial time evolution in the same manner as in the standard model, we take the activity patterns of the two networks. There is no main phase with extended rat trajectories and single neuron recordings.

## Simulation data analysis

### Spatial rate maps and autocorrelation functions

We discretize the animal's environment into $1\,\text{cm} \times 1\,\text{cm}$ position bins indexed by $\mathbf{R} = (X, Y)$. By tabulating a single neuron's average firing rate when the animal occupies each position, we produce the spatial rate map $S(\mathbf{R})$. We define its normalized spatial autocorrelation function as

$$C(\mathbf{R}) = \frac{\frac{1}{N(\mathbf{R})} \sum_{\mathbf{R}'} S(\mathbf{R}') S(\mathbf{R}' - \mathbf{R})}{\frac{1}{N(0)} \sum_{\mathbf{R}'} S(\mathbf{R}') S(\mathbf{R}')}, \quad (8)$$

where $N(\mathbf{R})$ is the number of pairs of positions separated by $\mathbf{R}$ can be efficiently calculated via discrete Fourier transforms.

We can define similar network autocorrelation functions $c(\mathbf{r})$ for the population activity within the neural sheet of each networks indexed $z$.

### Grid scale, orientation, and gridness

We use autocorrelation functions to extract the scale, orientation, and gridness of spatial and network grids. We first convert each position $\mathbf{R}$ to polar coordinates and calculate the autocorrelation as a function of radial distance $R$ by averaging over polar angle $\Phi$:

$$C_{rad}(R) = \frac{1}{N(R)} \sum_{\Phi} C(R, \Phi), \quad (9)$$

where $N(R)$ is the number of positions corresponding to each discretized $R$. This function is analogous to the radial distribution function of condensed matter physics. To filter out small fluctuations at the centimeter scale while permitting estimation of the location of extrema at the subcentimeter scale, we use coarse 1 cm bins for $C_{rad}(R)$, linearly interpolate its value at every 0.1 cm, and apply a Gaussian filter with respect to $R$ with standard deviation 8 cm. We

define the spatial grid scale $\Lambda$ as the $R$ corresponding to the first maximum of the smoothed $C_{rad}(R)$, not including the maximum at $R = 0$.

Grid orientation and gridness are computed from the angular structure of $C(R, \Phi)$ in the region around $R = \Lambda$. This region is an annulus bounded by $R$'s corresponding to the first and second minima of the smoothed $C_{rad}(R)$, which we call $R_1^*$ and $R_2^*$. This annulus is analogous to the first coordination shell of condensed matter physics. We average over $R$ within the annulus to calculate the autocorrelation as a function of $\Phi$:

$$C_{pol}(\Phi) = \frac{1}{N(\Phi)} \sum_{R_1^* \leq R \leq R_2^*} C(R, \Phi), \tag{10}$$

where $N(\Phi)$ is the number of positions within the annulus corresponding to each discretized $\Phi$. To assess the degree of sixfold symmetry, we calculate the sixth component of the discrete Fourier transform of $C_{pol}(\Phi)$ using 5° bins for $\Phi$:

$$\Psi_6 = \sum_{\Phi} C_{pol}(\Phi) e^{-6i\Phi}. \tag{11}$$

Orientation angle $\Theta$ is defined as the complex argument of $\Psi_6$ divided by 6. Gridness is the fraction of $C_{pol}(\Phi)$'s total Fourier power, after removing the zeroth component that describes its constant amplitude, belonging to the sixth component. It is thus

$$gridness = \frac{2|\Psi_6|^2}{\sum_{\Phi} C_{pol}(\Phi)^2 - \frac{1}{N_{pol}} \left[\sum_{\Phi} C_{pol}(\Phi)\right]^2}, \tag{12}$$

where $N_{pol} = 72$ is the number $\Phi$ bins. We need the factor of 2 to account for negative Fourier components which have power equal to that of positive components. By properties of Fourier transforms, $\sum_{\Phi} C_{pol}(\Phi)^2$ is the total Fourier power, and $[\sum_{\Phi} C_{pol}(\Phi)]^2 / N_{pol}$ is the power of the zeroth component. A similar definition for gridness has been proposed to assign a local grid score to each spike (**Weber and Sprekeler, 2019**). We use this definition instead of others used in the literature (**Stensola et al., 2012**) because it has an intuitive meaning as the fraction of angular power contributed by sixfold symmetry to the autocorrelation function.

Network grid scales $\lambda$, orientations $\theta$, and gridness can be similarly extracted via the network activity autocorrelation functions $c(\mathbf{r})$.

## Module clustering

Following (**Stensola et al., 2012**), we categorize grid cells into modules by clustering their grid scales and orientations using a $k$-means algorithm. The number of clusters $k$ is determined through kernel smoothed densities (KSDs).

We define linearly rescaled grid scales $\tilde{\Lambda}$ such that the largest and smallest scales for each simulation correspond to 0 and 1. We similarly define linearly rescaled grid orientations $\tilde{\Theta}$ such that 0° and 60° correspond to 0 and 1. We divide $\tilde{\Lambda}$-$\tilde{\Theta}$ space into $0.02 \times 0.02$ bins and define the KSD for each bin as

$$KSD(\tilde{\Lambda}, \tilde{\Theta}) = \frac{1}{N} \sum_i \exp\left[-\frac{(\tilde{\Lambda} - \tilde{\Lambda}_i)^2}{2\sigma_{\Lambda}^2}\right] \exp\left[-\frac{||\tilde{\Theta} - \tilde{\Theta}_i||^2}{2\sigma_{\Theta}^2}\right]. \tag{13}$$

$N$ is the number of grid cells, each of which has scale $\tilde{\Lambda}_i$ and $\tilde{\Theta}_i$. For the periodic variable $\tilde{\Theta}$, we define the distance $||c|| = |c|$ for $|c| \leq 0.5$, $1 - |c|$ for $|c| > 0.5$. We take both standard deviations $\sigma_{\Lambda}$ and $\sigma_{\Theta}$ to be 0.1. We use the number of peaks of this KSD as the initial number of clusters $k$ for $k$-means clustering in $\tilde{\Lambda}$-$\tilde{\Theta}$ space.

We perform $k$-means clustering with random initial points in $\tilde{\Lambda}$-$\tilde{\Theta}$ space 200 times per simulation. For each clustering attempt, we calculate the silhouette, a metric describing degree of separation among clusters (**Rousseeuw, 1987**; **Stensola et al., 2012**). For each grid cell $i$ in cluster $b$, we calculate its average distance in $\tilde{\Lambda}$-$\tilde{\Theta}$ space to all grid cells $j$ in cluster $c$:

$$D_{bi}^c = \frac{1}{N_c} \sum_j \sqrt{(\tilde{\Lambda}_{cj} - \tilde{\Lambda}_{bi})^2 + ||\tilde{\Theta}_{cj} - \tilde{\Theta}_{bi}||^2}, \tag{14}$$

where $N_c$ is the number of scales in cluster $c$. The silhouette of grid cell $i$ in cluster $b$ compares its average distance to other grid cells within its own cluster against its average distance to its closest cluster:

$$silhouette_{bi} = \frac{\min\limits_{c \neq b} D_{bi}^c - D_{bi}^b}{\max\left[\min\limits_{c \neq b} D_{bi}^c, D_{bi}^b\right]}. \tag{15}$$

The denominator is a normalization factor that rescales the silhouette between –1 and 1. More positive values indicate better clustering. Out of the 200 clustering attempts, we select the one with largest average silhouette across all grid cells. Finally, we reject all clusters with three or fewer grid cells from further analysis. The remaining clusters are grid cell modules.

## Extracting sheared triangular lattices for *Figure 7—figure supplement 1*

From vector graphics coordinates in *Stensola et al. (2012)* and *Krupic et al. (2015)*, we extract locations of the six autocorrelation peaks closest to the origin. These give us three lattice vectors. Distance is in arbitrary units.

We define the scale of each lattice to be the average length of the three lattice vectors. To obtain the orientation of each lattice, we calculate a circular mean of the angles of the lattice vectors. In contrast to the standard circular mean, this version has periodicity 60°:

$$\bar{\theta} = \frac{1}{6} \arctan \frac{\sum_i \sin 6\theta_i}{\sum_i \cos 6\theta_i}, \tag{16}$$

where $\theta_i$ is angle of each lattice vector and the arctangent accounts for the sign of the numerator and denominator of its argument.

These three lattice vectors $\mathbf{a}_i$ may not correspond to a perfect sheared grid, which is spanned by two independent lattice vectors. To evenly distribute the error introduced by the third lattice vector, we first choose the sign for each vector such that they are mutually separated by approximately 120°. We calculate reciprocal lattice vectors $\mathbf{b}_i$:

$$\mathbf{b}_i = \frac{2\pi \mathbf{a}_i^\perp}{|\mathbf{a}_i|^2}, \tag{17}$$

where $\mathbf{a}_i^\perp$ is the lattice vector $\mathbf{a}_i$ rotated by 90°. We then calculate the vector sum of these reciprocal lattice vectors, which should be the zero vector for a perfect grid in 2D. We subtract each original reciprocal lattice vector by a third of this sum to produce our corrected reciprocal lattice vectors $\mathbf{b}_i'$. Finally, we produce the lattice patterns $\rho(\mathbf{r})$ for the overlays in *Figure 7—figure supplement 1* by

$$\rho(\mathbf{r}) = \prod_i \frac{\cos(\mathbf{b}_i' \cdot \mathbf{r}) + 1}{2}. \tag{18}$$

## Appendix 2

DOI: https://doi.org/10.7554/eLife.46687.027

### Varying velocity gain model

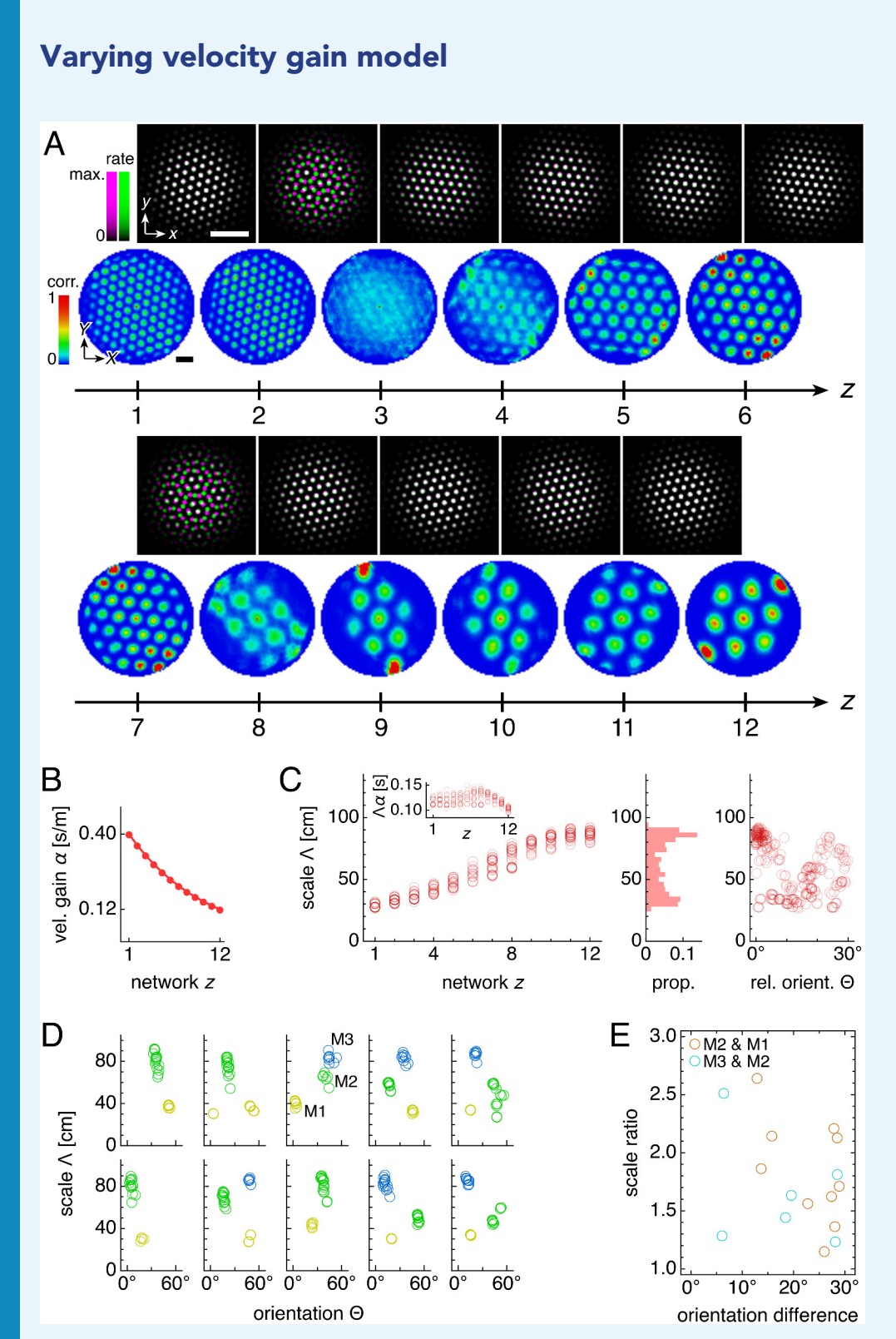

**Appendix 2—figure 1.** Simulations with a varying velocity gain α(z) and constant inhibition distance *l* produce modules that do not exhibit preferred relationships. (**A**) Representative

simulation. Top rows: activity overlays between adjacent networks with the network at smaller (larger) $z$ depicted in magenta (green). Bottom rows: spatial autocorrelations of spatial rate maps. (**B**) Velocity gain profile $\alpha(z)$. (**C–E**) Data from 10 replicate simulations. (**C**) Left: spatial grid scales $\Lambda(z)$. For each network, there are up to 30 red circles corresponding to three neurons recorded during each simulation. Inset: $\Lambda(z)$ multiplied by the velocity gain $\alpha(z)$. Middle: histogram for $\Lambda(z)$ collected across all networks. Right: spatial grid orientations $\Theta$ relative to the grid cell in the same simulation with largest scale. (**D**) Distributions of spatial grid scales and orientations for each replicate simulation. Due to hexagonal symmetry, orientation is a periodic variable modulo 60°. Different colors indicate separate modules. The ninth panel corresponds to the overlays in **A**. (**E**) Spatial grid scale ratios and orientation differences between adjacent modules. Maximum velocity gain $\alpha_{max} = 0.40$ s/m, minimum velocity gain $\alpha_{min} = 0.12$ s/m, and scaling exponent $\alpha_{exp} = 0$. Network size $n \times n = 174 \times 174$, coupling spread $d = 12$, coupling strength $u_{mag} = 0.6$, and inhibition distance $l = 6$. Other parameter values are in **Table 1**. White scale bars, 50 neurons. Black scale bars, 50 cm.

DOI: https://doi.org/10.7554/eLife.46687.029

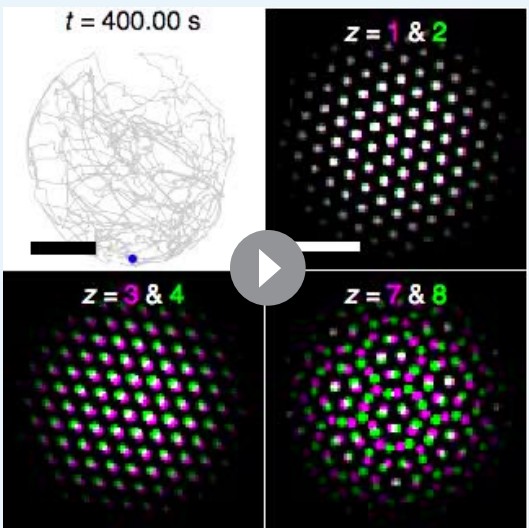

**Appendix 2—video 1.** Last 100 s of the simulation displayed in **Appendix 2—figure 1A**. Top left: accumulated rat trajectory (gray curve) with current rat position (blue dot). Top right, bottom left, and bottom right: activity overlays between adjacent networks with the network at smaller (larger) $z$ depicted in magenta (green), so white indicates regions of activity in both networks. White scale bar, 50 neurons. Black scale bar, 50 cm.

DOI: https://doi.org/10.7554/eLife.46687.030

## Simulation setup with a velocity gain gradient

These simulations use constant inhibition distance $l$ and a varying velocity gain $\alpha(z)$ (**Appendix 2—figure 1B**). The functional form for $\alpha(z)$ is similar to that for $l(z)$ of the inhibition gradient model (see Materials and methods), except it decreases with $z$ instead of increasing:

$$\alpha(z) = \left[ \alpha_{max}^{\alpha_{exp}} + \left( \alpha_{min}^{\alpha_{exp}} - \alpha_{max}^{\alpha_{exp}} \right) \frac{z-1}{h-1} \right]^{1/\alpha_{exp}}, \tag{19}$$

which ranges from $\alpha_{max} = \alpha(1)$ to $\alpha_{min} = \alpha(h)$ with concavity tuned by $\alpha_{exp}$. More negative values of $\alpha_{exp}$ lead to greater concavity; for $\alpha_{exp} = 0$, we use the limiting expression $\alpha(z) = \alpha_{max}^{(h-z)/(h-1)} \alpha_{min}^{(z-1)/(h-1)}$.

Simulation initialization and time evolution proceed similarly to the inhibition gradient model, except we evolve 250000 timesteps with a real rat trajectory before starting the main simulation phase, instead of 50000 timesteps. Simulations with a velocity gain gradient tend to

have transient configurations that persist longer before changing to a stable configuration, so a longer initialization period helps the main simulation start in a stable configuration.

## Results with a velocity gain gradient

Simulations with a velocity gain gradient and excitatory coupling exhibit modularity, but grid scale and orientation relationships vary greatly among replicate simulations that use different random initial firing rates. Single neuron autocorrelation maps in *Appendix 2—figure 1A* show that this model can produce a grid system with range of grid scales. Note that the population activity contains grids of the same scale for all networks because the inhibition distance is constant. Spatial scales are smaller at lower $z$ because they have higher velocity gain α (*Appendix 2—figure 1B*) and translate their activity patterns in proportion to rat motion more rapidly (see also *Appendix 2—figure 2*). Plotting the spatial scales and orientations of all replicate simulations does not reveal strong clustering (*Appendix 2—figure 1C*), but separate analysis of each replicate simulation allows us to identify mostly well-defined modules (*Appendix 2—figure 1D*). However, scale ratios and orientation differences between adjacent modules do not cluster around preferred values (*Appendix 2—figure 1E*).

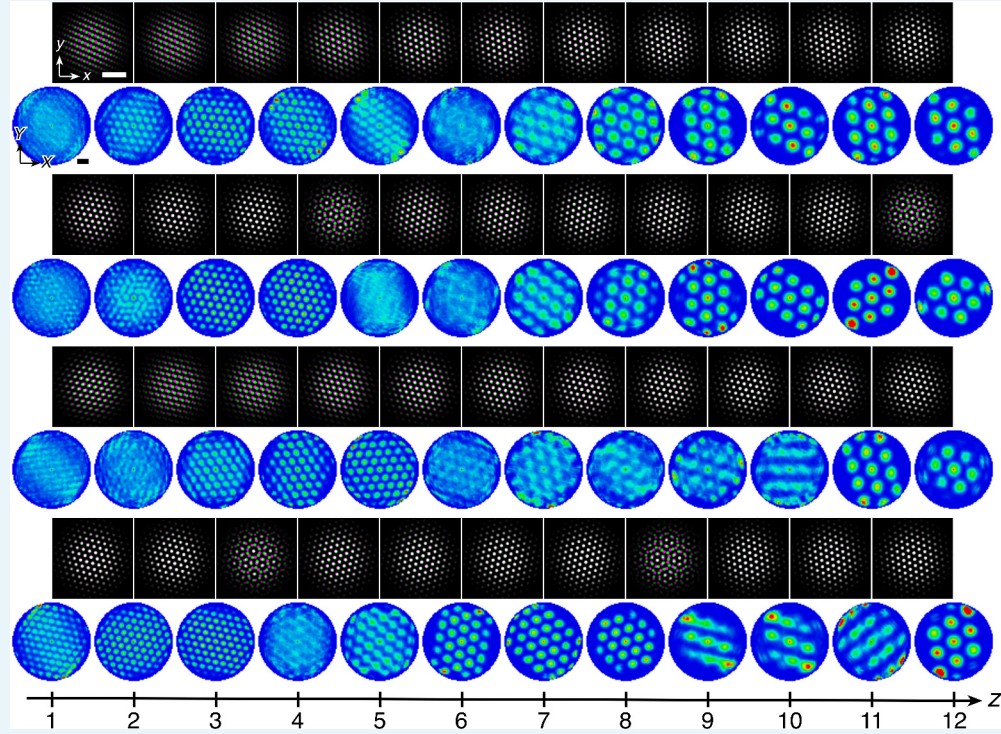

**Appendix 2—figure 2.** Activities of additional replicate simulations with a varying velocity gain contributing to *Appendix 2—figure 1*. Top rows: activity overlays between adjacent networks with the network at smaller (larger) $z$ depicted in magenta (green). Bottom rows: autocorrelations of spatial rate maps. White scale bars, 50 neurons. Black scale bars, 50 cm.
DOI: https://doi.org/10.7554/eLife.46687.031

To further investigate the dynamics of this model, we follow three pairs of adjacent networks in *Appendix 2—Video 1* which corresponds to the replicate simulation shown in *Appendix 2—figure 1A*. The movie depicts the activity overlays of these networks as the simulated rat explores its enclosure. We first consider the overlay between networks 1 and 2. Due to the 'rigidity' provided by excitatory coupling as described in the main text, the population activities of these two networks remain in registry throughout the movie; thus, grid cells from these two networks have the same spatial scale and orientation and belong to the same module. Now consider the overlay between networks 3 and 4. For most of the movie,

their population activities are in registry. However, the gradient in velocity gain prefers the more dorsal network (smaller $z$) to have an activity pattern that translates more rapidly. This effect can disrupt the rigidity imposed by coupling and, at $t \approx 470$ s, one pattern jumps along a lattice vector relative to the other. Such an anomaly implies that at least one of the two networks cannot have an activity pattern that translates proportionally to the entire rat trajectory, a requirement for faithful path-integration. Indeed, the spatial autocorrelation map for $z = 3$ in *Appendix 2—figure 1A* shows a lack of grid-like symmetry. This example illustrates how a velocity gain gradient can disrupt grid cells in a way that an inhibition distance gradient does not—the latter does not resist the rigidity of excitatory coupling through different translation speeds of activity patterns.

Finally, consider the overlay between networks 7 and 8 in *Appendix 2—Video 1*. Here, the two activity patterns remain rotated relative to each other, with little registry. Coupling causes the activity peaks of network 8 to preferentially excite the corresponding areas of network 7, but since there are few peaks in those areas, the effect of coupling is weak between these two networks. Therefore, the activity patterns can freely glide relative to each other, each translating proportionally with animal motion but with different speeds preferred by different velocity gains. Indeed, single neuron spatial rate maps for $z = 7$ and 8 show different scales (*Appendix 2—figure 1A*), which identifies this lack of registry as a mechanism for producing interfaces between grid modules. However, this mechanism does not enforce how quickly one pattern glides relative to the other and thus does not lead to preferred scale ratios (*Appendix 2—figure 1E*). It does require that activity patterns stay rotated relative to each other, which may explain the abundance of large orientation differences >15° between modules (*Appendix 2—figure 1E*).

Thus, excitatory coupling with a velocity gain gradient can produce grid modules, but, in contrast to the model with varying inhibition distance, the velocity gain gradient model does not favor certain scale ratios and orientation differences. Coupling between attractor networks with different velocity gains may perform a different role: it can make path-integration more robust against input noise (*Mosheiff and Burak, 2019*).

## Appendix 3

DOI: https://doi.org/10.7554/eLife.46687.027

## Quasicrystal approximant grids

Within certain parameter ranges, the coupled system can give rise to quasicrystal approximant grids. One example simulation with dorsal-to-ventral coupling is shown in *Appendix 3–Figure 1*. From $z = 6$ to 9, network activity peaks form the vertices of a square-triangle tiling that is a dodecagonal quasicrystal approximant (*Stampfli, 1986*; *Levine and Steinhardt, 1986*). This tiling is labeled $(3^6; 3^2.4.3.4)$ based on the type and order of regular polygons that meet at its vertices (*Grünbaum and Shephard, 1977*). The $z = 8$ and 9 Fourier power spectra approach 12-fold symmetry, as expected from a dodecahedral quasicrystal approximant. From $z = 10$ to 12, the network activity patterns demonstrate twofold dihedral symmetry.

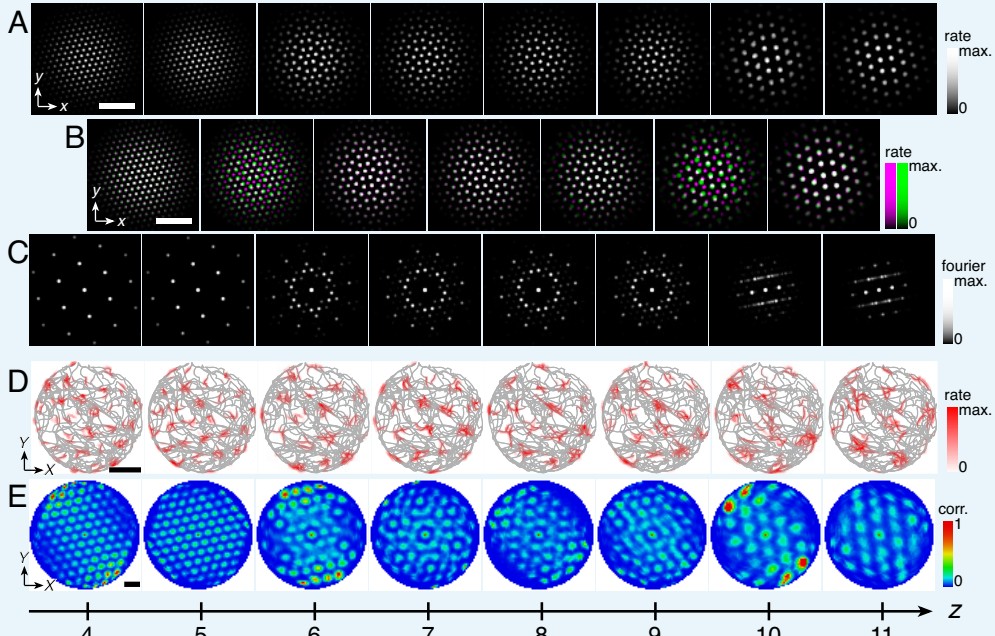

**Appendix 3—figure 1.** Quasicrystal approximant grids in a simulation with dorsal-to-ventral coupling. Networks $z = 4$ to 11 shown out of 12 total. (**A**) Network activities at the end of the simulation. (**B**) Activity overlays between adjacent networks depicted in the top row. In each panel, the network at smaller (larger) $z$ is depicted in magenta (green), so white indicates regions of activity in both networks. (**C**) Fourier power spectra for network activities with the origin at the center of each image and the edges cropped. (**D**) Spatial rate map of a single neuron for each $z$ superimposed on the animal's trajectory. (**E**) Spatial autocorrelations of the rate maps depicted in **D**. Network size $n \times n = 230 \times 230$, coupling spread $d = 2$, coupling strength $u_{mag} = 0.6$, maximum inhibition distance $\alpha_{exp} = 0$, and velocity gain $\alpha = 0.12$ s/m. Other parameter values are in *Table 1*. White scale bars, 50 neurons. Black scale bars, 50 cm.
DOI: https://doi.org/10.7554/eLife.46687.033

