## [Decision Letter]

Thank you for submitting your article "A geometric attractor mechanism for self-organization of entorhinal grid modules" for consideration by *eLife*. Your article has been reviewed by three peer reviewers, and the evaluation has been overseen by a Reviewing Editor and Laura Colgin as the Senior Editor. The following individual involved in review of your submission has agreed to reveal their identity: Yi Gu (Reviewer #2).

The reviewers have discussed the reviews with one another and the Reviewing Editor has drafted this decision to help you prepare a revised submission.

Summary:

This study presents a model for MEC grid module formation emerging from a series of attractor networks (based on the Burak-Fiete model) coupled by excitatory interactions. The coupling causes the length-scales of each network to cluster into discrete bands, reminiscent of experimental modules of increasing scale. The orientation of the networks also cluster, again consistent with experiment. Thus a rather simple addition to the network model architecture brings it into agreement with a large body of experiments.

Essential revisions:

1) The authors must provide more links from model predictions to experiments. The reviewers have made some suggestions both to link the model to existing data, and for new predictions, and the authors could of course come up with further such links.

2) The authors must examine the assumption of precise positional projections between networks. We would prefer to see some test simulations that explore imprecision in this quantity.

3) The authors should address some of the other key parameters of the simulations, particularly the network size, noise, spiking dynamics, and plasticity. The reviewers would prefer to see simulations for at least some of these, though we understand that others may be a substantial effort and the authors can address those in the Discussion.

*Reviewer #1:*

This study presents a model for MEC grid module formation emerging from a series of attractor networks (based on the Burak-Fiete model) coupled by excitatory interactions. The coupling causes the length-scales of each network to cluster into discrete bands, reminiscent of experimental modules of increasing scale. The orientation of the networks also cluster, again consistent with experiment. Thus a rather simple addition to the network model architecture brings it into agreement with a large body of experiments. Some aspects of these findings have been seen in models proposed by Treves and co-workers, but the authors point out that there are features such as orientation, that only the current model predicts.

1) The model makes several predictions, including the modules, their orientation, firing rate variability. These all emerge from this rather parsimonious elaboration to the attractor network model. This is a strong point of this study.

2) A key requirement for these outcomes is the coupling between networks, which requires a certain precision of connectivity between successive networks. The authors have made predictions for the outcome of lesion experiments, but I would like to ask if there are any more direct projection or connectivity studies that support this proposed circuitry. They mention some studies involving recurrent connectivity among grid cells, but it isn't clear to me that these studies demonstrate the spatial structure that the current model requires.

3) Another possible and testable manipulation would be to make a small focal lesion rather than a sub-network wide one. It would be interesting to see how this affects the z<lesion networks, and this might provide a more stringent and nuanced prediction for experiments.

4) In an ideal world one would like to compare predictions for specific manipulations between the current model and others in the field. I would specifically be interested in seeing if there are manipulations which would strongly contrast the properties of the current model and that of Treves and co-workers.

5) The authors explore a range of simulation parameters, notably coupling strengths and the ratio of inhibition distances. However, they barely touch on spiking dynamics and plasticity in a line in the Discussion. I therefore get a sense that the model has been sensitivity tested only along a very few dimensions. It would be reassuring to see somewhat more exploration of these properties, especially those that relate to more biological realism.

*Reviewer #2:*

This paper proposed a mechanism for generating discrete grid modules (Stensola et al., 2012) in attractor networks of medial entorhinal cortex (MEC) by combining lateral inhibition within individual "networks" and excitatory interactions between networks. Modulating the balance between the inhibition and excitation led to constant scale ratios and orientation differences between adjacent modules, which were consistent with experimental data. This paper is very well written and this first demonstration of a potential mechanism for generating grid modules in attractor networks of the MEC would be of high interest to neuroscience readers. However, providing additional connections between the proposed theory and experimental observations would make this work more significant.

1) Numbers of neurons in each module: the current theory was developed based on 12 "attractor networks" the MEC. Each network contained 160×160 neurons and these 307,200 grid cells mostly gave three modules. In reality, an animal could have four or five modules (Stensola et al., 2012), so there might be even a larger number of grid cells per animal based on the theory. Given the fact that grid cell population is only ~20% (or even less, ~5% in Miao C et al., Cell, 2017) of the MEC cells, and the relatively low number of grid cells per module recorded by tetrodes and imaging (Stensola et al., 2012; Gu et al., 2018), this theoretical number of grid cell seems unrealistically large. Although it would be hard to know exactly how many grid cells are in real animals, the question is how the conclusions are sensitive to the size of the network. For a network contains half or even a quarter of neurons used here (less number of neurons per network or less number of networks), are these conclusions (the coupled excitation and lateral inhibition generate constant scale ratio and orientation differences between adjacent modules) still true?

2) Neurons coupled by excitation: the theory is developed under the assumption that a neuron at a given position of a ventral network excited the neuron at the same position of a dorsal network (Figure 1D, bidirectionally if in Figure 3C). Thinking about the noise of real network connectivity, how tolerant is this theory to the disruption of this position correspondence of the excitatory connection across networks?

3) Variation of grid field amplitudes: the authors claimed that the excitation from the ventral to dorsal modules could lead to the variation in grid field amplitudes for cells in dorsal modules, as observed experimentally (Ismakov, 2017; Dunn, 2017). However, this statement is rather weak. Based on the theory, for a dorsal grid cell, its fields, which aligned with the fields of a ventral grid cell that excited this dorsal cell, should have higher amplitudes. The amplitude variation of grid fields should have a particular pattern (Figures 2B and 4D). It would be helpful to see more specific explanations for real examples of grid cell activity based on the current theory, i.e. what commensurate or discommensurate lattices could be responsible for generating a given pattern of grid field amplitudes and under what kind of excitation and inhibition.

4) Discommensurate lattices for real grid modules: the author claimed that discommensurate lattice relationships could produce realistic modules (Figure 5). Similar to (3), this statement would be more convincing if the author could give several examples of adjacent modules recorded from the same animal and explain the discommensurate lattices and the detailed parameters of excitation and inhibitions (strength and spread of excitation, and ratio of inhibition distances between modules) that used to form these modules.

5) Independent rescaling of grid modules in different environments: previous work showed that grid scales of different modules could change independently when an environment was deformed (Stensola et al., 2012, Figure 7). However, based the current theory, the scale ratio of adjacent modules seemed to be constant, unless the balance between the excitation and inhibition is changed. How could the current theory explain the independent rescaling of different modules? This question could also be in line with the last sentence in the "Discussion" about border cells and environmental deformation. In general, can the author expand this discussion by speculating the mechanism for the change of grid scales (maybe orientations too) in different modules in different (or deformed) environments and how border cells play roles in this process (i.e. how do border cells interfere with the balance of excitation and inhibition)?

*Reviewer #3:*

This paper is quite well written and comprehensive. It addresses an important question, namely, what are the mechanisms responsible for the modular organization of grid cells? In doing so it arrives at some general principles of network organization in the MEC. Overall, I think it needs no major changes.

An earlier paper by one of the authors showed that grid cell modules are arranged in a manner that minimizes the number of neurons required to encode location with a given resolution. In this paper, they look at how such a peculiar modular organization emerges in a model attractor network. To construct individual modules the authors used a well-known continuous attractor network by Burak and Fiete. The grid scale is determined by the spatial extent of inhibition in this network. The authors connected a set of 12 such attractor networks with a gradually varying grid scale using excitatory connections across neighboring networks. The observed spatial scale of grid cell receptive fields in each attractor network did not follow the gradual increase that would be the case if they were uncoupled, but clustered into groups, with the scale ratios across groups matching experimental observations.

This paper addresses an important question and does so using an innovative and simple extension of an existing model. The manuscript is clearly written, potential caveats have been addressed, and the figures are detailed (albeit tiny). The authors arrive at an intuitive explanation for the location of fractures where the grid cell receptive fields transition from one scale to the next. Given the complex dynamics of stellate cells and pyramidal cells in layer II, it is quite surprising that the patterns that emerge from this phenomenological model can be quantitatively compared to the results of experiments. What is particularly interesting is that the model is difficult to break, in that different excitatory connectivities (bi-directional and uni-directional in either direction), all seem to generate the same modularity ratios. It seems like the model hints at general principles at work in the MEC that the authors allude to in the Discussion.

The authors mention that at the boundaries an attractor network can be part of one module or the other depending on the initial conditions. Here it would be useful to understand whether a grid-like receptive field persists when temporal noise is added to the system.

---

## [Author Response]

Essential revisions:1) The authors must provide more links from model predictions to experiments. The reviewers have made some suggestions both to link the model to existing data, and for new predictions, and the authors could of course come up with further such links.

We have added further connections between model predictions and experiments in three ways. In Figure 8—figure supplement 1, we provide predictions for additional lesion protocols, as suggested by reviewer 1. In Figure 7—figure supplement 2, we provide examples for how module relationships in experimental recordings may arise from lattice relationships predicted by our model, as suggested by reviewer 2. In Figure 6—figure supplement 2, we provide an example for how structured field-to-field firing rate variability in an experimental recording may arise from a discommensurate lattice relationship predicted by our model, as suggested by reviewer 2. These additions will strengthen the interpretability and the predictive power of our model.

2) The authors must examine the assumption of precise positional projectionsbetween networks. We would prefer to see some test simulations that exploreimprecision in this quantity.

Figure 4 and Figure 5—figure supplement 3 now provide results that demonstrate robustness of our results to variations in directionality, positional spread, and positional noise in the excitatory coupling between networks.

3) The authors should address some of the other key parameters of the simulations, particularly the network size, noise, spiking dynamics, and plasticity. The reviewers would prefer to see simulations for at least some of these, though we understand that others may be a substantial effort and the authors can address those in the Discussion.

Figure 4 and Figure 5—figure supplement 3 now provide results for systems with smaller network size, temporal and coupling noise, and spiking dynamics. Implementing plasticity in our model would be a substantial effort, and so, as the Editor suggests, and we address its possible effects in the Discussion.

Reviewer #1:1) The model makes several predictions, including the modules, theirorientation, firing rate variability. These all emerge from this ratherparsimonious elaboration to the attractor network model. This is a strongpoint of this study.

Thank you for this assessment. We were very pleased to find that such a simple extension to the attractor network model seems to account for a number of experimental findings.

2) A key requirement for these outcomes is the coupling between networks, which requires a certain precision of connectivity between successive networks.The authors have made predictions for the outcome of lesion experiments, butI would like to ask if there are any more direct projection or connectivity studies that support this proposed circuitry. They mention somestudies involving recurrent connectivity among grid cells, but it isn't clearto me that these studies demonstrate the spatial structure that the currentmodel requires.

In the Discussion, we have elaborated upon the connectivity studies reported in the literature that find excitatory connections among superficial layers of the MEC. In short, thus far these studies have found short-range connections and very long-range connections across hemispheres. Our model predicts excitatory connections between locations along the MEC corresponding to different modules. An observation of such connections would support our model.

3) Another possible and testable manipulation would be to make a small focallesion rather than a sub-network wide one. It would be interesting to see howthis affects the z<lesion networks, and this might provide a more stringentand nuanced prediction for experiments.

We thank the reviewer for this excellent suggestion. We now provide predictions for such a regional lesion in Figure 8—figure supplement 1 and in the Results section “Testing for coupling with a mock lesion experiment”. The figure supplement also contains predictions for a global lesion that spares one neuron in every 3 x 3 block of the lesioned network.

4) In an ideal world one would like to compare predictions for specificmanipulations between the current model and others in the field. I wouldspecifically be interested in seeing if there are manipulations which wouldstrongly contrast the properties of the current model and that ofTreves and co-workers.

We agree with the reviewer that our model and that of Treves and co-workers (Urdapilleta et al., 2017) should be distinguishable by experimental tests. In the first place, we are studying an attractor model in which grids form through a collective effect of the interactions in a network. The Urdapilleta et al. paper uses a firing rate adaptation model which generates grids through a fundamentally different mechanism: cells with different time constants produce grids of different scales. Thus, fundamentally, we need experiments testing whether grids are formed by a collective spatial attractor mechanism or through a temporal single-cell firing rate adaption mechanism.

Urdapilleta et al. extend the firing rate adaptation model for grid cells by adding excitatory coupling among these cells of different scales. This causes clustering in scales and orientations but, unlike our model, does not have a mechanism to dynamically enforce the average constancy of grid scale ratios, which appears to be a feature of the grid system (Barry et al., 2007; Stensola et al., 2012; Krupic et al., 2015). We state this in the Discussion.

We believe that the two models can be most effectively differentiated by careful measurements of the orientation differences between modules in intact animals. We have now emphasized in the Discussion that our model allows for orientation differences that are significantly different from zero as sometimes seen in, e.g., Krupic et al., 2015. In contrast, Treves and co-workers report orientation differences that are all within one standard deviation away from zero (Table 1 of Urdapilleta, et al., 2017).

5) The authors explore a range of simulation parameters, notably couplingstrengths and the ratio of inhibition distances. However, they barely touchon spiking dynamics and plasticity in a line in the Discussion. I thereforeget a sense that the model has been sensitivity tested only along a very fewdimensions. It would be reassuring to see somewhat more exploration of theseproperties, especially those that relate to more biological realism.

We have now tested robustness of our results to variations in directionality, positional spread, and positional noise in the excitatory coupling between networks We also tested robustness for systems with smaller network size, temporal and coupling noise, and spiking dynamics. These new tests appear in Figure 4 and Figure 5—figure supplement 3. Implementing plasticity in our model would be a substantial additional effort and is out of the scope of this manuscript; so we addressed its possible effects in the Discussion.

Reviewer #2:1) Numbers of neurons in each module: the current theory was developed based on 12 "attractor networks" the MEC. Each network contained 160×160 neurons and these 307,200 grid cells mostly gave three modules. In reality, an animal could have four or five modules (Stensola et al., 2012), so there might be even a larger number of grid cells per animal based on the theory. Given the fact that grid cell population is only ~20% (or even less, ~5% in Miao C et al., Cell, 2017) of the MEC cells, and the relatively low number of grid cells per module recorded by tetrodes and imaging (Stensola et al., 2012; Gu et al., 2018), this theoretical number of grid cell seems unrealistically large. Although it would be hard to know exactly how many grid cells are in real animals, the question is how the conclusions are sensitive to the size of the network. For a network contains half or even a quarter of neurons used here (less number of neurons per network or less number of networks), are these conclusions (the coupled excitation and lateral inhibition generate constant scale ratio and orientation differences between adjacent modules) still true?

The reviewer suggests a good opportunity to exhibit the robustness of our model. Our main simulation uses a large number of neurons in each network to clearly illustrate the geometric relationships between attractor bumps. It uses a large number of networks to demonstrate that our model can produce modules with grid scales that jump sharply, even when the incremental changes in inhibition distances are small from one network to the next. We now show results for systems with 11% of the original number of neurons in Figure 4F. It contains approximately 35,000 neurons and forms 3 modules.

2) Neurons coupled by excitation: the theory is developed under the assumption that a neuron at a given position of a ventral network excited the neuron at the same position of a dorsal network (Figure 1D, bidirectionally if in Figure 3C). Thinking about the noise of real network connectivity, how tolerant is this theory to the disruption of this position correspondence of the excitatory connection across networks?

The suggestion of coupling noise is another good opportunity to exhibit the robustness of our model. In Figure 4 and Figure 5—figure supplement 3 we show that our model is robust to such positional noise in the excitatory coupling.

3) Variation of grid field amplitudes: the authors claimed that the excitation from the ventral to dorsal modules could lead to the variation in grid field amplitudes for cells in dorsal modules, as observed experimentally (Ismakov, 2017; Dunn, 2017). However, this statement is rather weak. Based on the theory, for a dorsal grid cell, its fields, which aligned with the fields of a ventral grid cell that excited this dorsal cell, should have higher amplitudes. The amplitude variation of grid fields should have a particular pattern (Figures 2B and 4D). It would be helpful to see more specific explanations for real examples of grid cell activity based on the current theory, i.e. what commensurate or discommensurate lattices could be responsible for generating a given pattern of grid field amplitudes and under what kind of excitation and inhibition.

We are grateful to the reviewer for this suggestion to illustrate a sample connection between our model and experimental data. In Figure 6—figure supplement 2, we provide an example comparison between a pattern of firing rates in a recorded neuron (Dunn et al., 2017) and a simulated neuron that participates in a discommensurate relationship. We caution that proper testing of our predictions requires a comprehensive analysis with much more data, preferably with grid cells recorded from a circular environment to prevent confounding effects from environmental boundaries.

4) Discommensurate lattices for real grid modules: the author claimed that discommensurate lattice relationships could produce realistic modules (Figure 5). Similar to (3), this statement would be more convincing if the author could give several examples of adjacent modules recorded from the same animal and explain the discommensurate lattices and the detailed parameters of excitation and inhibitions (strength and spread of excitation, and ratio of inhibition distances between modules) that used to form these modules.

We are similarly grateful for this suggestion. In Figure 7—figure supplement 2, we provide an example for how a series of experimentally recorded grid cells from different modules can arise from various lattice relationships. Again, we caution that a detailed test requires a comprehensive analysis with more data, preferably with grid cells recorded from a circular environment to prevent confounding effects from environmental boundaries, or with an extension of the theory include effects on grid orientation of anchoring to boundaries (see, e.g., Keinath et al., 2018).

5) Independent rescaling of grid modules in different environments: previous work showed that grid scales of different modules could change independently when an environment was deformed (Stensola et al., 2012, Figure 7). However, based the current theory, the scale ratio of adjacent modules seemed to be constant, unless the balance between the excitation and inhibition is changed. How could the current theory explain the independent rescaling of different modules? This question could also be in line with the last sentence in the "Discussion" about border cells and environmental deformation. In general, can the author expand this discussion by speculating the mechanism for the change of grid scales (maybe orientations too) in different modules in different (or deformed) environments and how border cells play roles in this process (i.e. how do border cells interfere with the balance of excitation and inhibition)?

The interpretation of the experimental rescaling data is complicated, with the original explanation of rescaling contested by data analysis that shows direction-dependent field shifts instead (Keinath et al., 2018). In the latter interpretation, phase relationships with border cells that are learned during familiarization with an environment produce trajectory dependent grid phase shifts when the environment is deformed. According to this interpretation, and the evidence shown in Keinath et al., averaging the data over time produces the appearance of rescaling, but in fact the grids are simply shifting left, right, up or down depending on the last contacted wall in the deformed environment. The appearance of rescaling in the time-averaged fields is simply more prominent in the large grids, and less so in the small ones. Meanwhile, conditioning the data on the last encounter with a boundary leads to grids that do not rescale and maintain their scale ratios. Thus, following Keinath et al. and results from the Giocomo lab, we are not convinced that there is independent rescaling of grid modules. Including the effects of environmental boundaries is beyond the scope of the current work and thus we leave investigation of the effects of boundary deformations to future authors. We have expanded on this in the Discussion.

Reviewer #3:[…] The authors mention that at the boundaries an attractor network can be part of one module or the other depending on the initial conditions. Here it would be useful to understand whether a grid-like receptive field persists when temporal noise is added to the system.

We are grateful for this opportunity to demonstrate the robustness of our model, and Figure 4G and Figure 5—figure supplement 3 now include simulations with temporal noise and demonstrate the persistence of precise modules.